# Coresets for Mixtures of (arbitrarily large) Gaussians

## Abstract

An $\varepsilon$-coreset for $k$-Gaussian Mixture Models ($k$-GMMs) of an input set $P \subseteq \mathbb{R}^d$ of points, is a small weighted set $C \subset P$, such that the negative log-likelihood $L(P, \theta)$ of every $k$-GMM $\theta$ is provably approximated by $L(C, \theta)$, up to a multiplicative factor of $1 \pm \varepsilon$, for a given $\varepsilon > 0$. Existing coreset [NIPS'11, JMLR'18] approximates only "semi-spherical" $k$-GMMs, whose covariance matrices are similar to the identity matrix. This work provides the first randomized algorithm that computes a coreset for arbitrary large $k$-GMMs. This is by forging new links to projective clustering and modern techniques in computational geometry. Experimental results on real-world datasets that demonstrate the efficacy of our approach is also provided.

## 1 Introduction

This paper suggests a coreset for the fundamental machine learning problem of fitting a given set of $n$ inputs points (the dataset) by Gaussian mixture model. In this section, we formally define this model, and a coreset that approximates its input dataset for any given model. Due to space constraints, most of the technical parts, and discussions, including most of the proofs and pseudo-codes, appear at the appendix, while the main text serves as its summary.

**Gaussian mixture models.** A *multivariate Gaussian*$(\Sigma, \mu)$ in the $d$-dimensional Euclidean space $\mathbb{R}^d$ consists of a positive definite matrix $\Sigma \in \mathbb{R}^{d \times d}$, called *covariance matrix*, and a *mean* $\mu \in \mathbb{R}^d$. The probability density function (pdf) of such a multivariate Gaussian (or Normal) distribution at a data point $p \in \mathbb{R}^d$ is

$$\mathcal{N}(p \mid \mu, \Sigma) := \frac{1}{\sqrt{\det(2\pi\Sigma)}} \exp\left(-\frac{1}{2}(p - \mu)^T \Sigma^{-1}(p - \mu)\right),$$

where $\det(\cdot)$ denotes the determinant of a matrix, and $\exp(x) := e^x$ for $x \in \mathbb{R}$.

A $k$-*Gaussian mixture model* ($k$-GMM) $\theta$ is a distribution $\omega \in [0, 1]^k$ over a set ("mixture") of such $k$ Gaussians. Formally, $k \geq 1$ is an integer, $\sum_{i=1}^{k} \omega_i = 1$, and $\theta$ is a set that consists of $k$ three tuples

$$\theta := \left\{(\omega_i, \Sigma_i, \mu_i)\right\}_{i=1}^{k} := \left\{(\omega_1, \mu_1, \Sigma_1), \cdots, (\omega_k, \mu_k, \Sigma_k)\right\}, \tag{1}$$

where $\mu_i \in \mathbb{R}^d$ is the mean and $\Sigma_i \in \mathbb{R}^{d \times d}$ is the covariance matrix of the $i$th Gaussian, for every $i \in [k] := \{1, \cdots, k\}$. The GMM *likelihood* that a given point $p \in \mathbb{R}^d$ was generated by a $k$-GMM $\theta$ is the pdf

$$\Pr(p \mid \theta) := \sum_{i=1}^{k} \omega_i \mathcal{N}(p \mid \mu_i, \Sigma_k).$$

The likelihood that a given non-empty finite set $P \subseteq \mathbb{R}^d$, was generated by $\theta$ is thus

$$\Pr(P \mid \theta) := \prod_{p \in P} \Pr(p \mid \theta) = \prod_{p \in P} \sum_{i=1}^{k} \omega_i \mathcal{N}(p \mid \mu_i, \Sigma_k)$$

$$= \prod_{p \in P} \sum_{i=1}^{k} \omega_i \frac{1}{\sqrt{\det(2\pi\Sigma_i)}} \exp\left(-\frac{1}{2}(p - \mu_i)^T \Sigma_i^{-1}(p - \mu_i)\right). \tag{2}$$

In a *GMM fitting problem*, the input is an integer $k \geqslant 1$ and a dataset $P \subseteq \mathbb{R}^d$. The goal is to compute a $k$-GMM $\theta$ that maximizes the likelihood in equation 2, possibly together with constraints, or, more generally, *regularization function* $\lambda : \Theta \to \mathbb{R}$ that depends only on $\theta$ Schölkopf et al. (2002) and incorporates prior structural knowledge. For example, the sparsity of $\theta$ Fop et al. (2019), the ratio between its eiganvalues, restricted zones or means Qiao and Li (2015), or in a deep generative approach Manduchi et al. (2021). Since the likelihood involves multiplying many small probability density values, the result can become extremely small, leading to numerical underflow issues. Therefore, it is more common and practical to work instead with the *negative log-likelihood*

$$L(P, \theta) := -\ln \Pr(P \mid \theta) = -\sum_{p \in P} \ln \sum_{i=1}^{k} \frac{\omega_i}{\sqrt{\det(2\pi\Sigma_i)}} \exp\left(-\frac{1}{2}(p - \mu_i)^T \Sigma_i^{-1}(p - \mu_i)\right). \quad (3)$$

A $k$-GMM $\theta$ that maximizes equation 2 with a regularization function then becomes

$$\min_{\theta \in \Theta_{d,k}} L(P, \theta) + \lambda(\theta), \quad (4)$$

where $\Theta_{d,k}$ is the (possibly infinite) set of feasible $k$-GMMs, which may capture arbitrary constraints for the fitting problem at hand.

**The set $\Theta_{d,k}$.** We assume that all the eigenvalues of the inverse covariance matrices $\Sigma_1^{-1}, \cdots, \Sigma_k^{-1}$ of every $k$-GMM $\theta \in \Theta_{d,k}$ are at least 1. Otherwise, we multiply them by the same scaling factor (which is independent of the input $P$), and update $\lambda(\theta)$ accordingly; see Lemma C.1 and Feldman et al. (2011); Lucic et al. (2017). Such a lower bound on the eigenvalues is essential to avoid division by zero in equation 3 and degenerate solutions by taking arbitrarily small eigenvalues; see discussions e.g. in García-Escudero et al. (2018); Rocci et al. (2018); Ingrassia (2004) and implementations details in Shental et al. (2003); Athey et al. (2019). However, unlike Feldman et al. (2011); Lucic et al. (2017), no upper bound is assumed on these eigenvalues or their mutual ratios.

**Coresets.** A possible approach for solving the $k$-GMM fitting problem and its many variants above, is to develop new algorithms from scratch, or improve existing heuristics such as the common EM-algorithm (Expected Maximization) that is used to find a local optimum Shental et al. (2003); Athey et al. (2019). Instead, we suggest *coresets* for this and related problems. In this paper, a coreset for a given finite input set $P$ of points is a *subset* $C \subseteq P$ with a ("weight") function $w : C \to [0, \infty)$, such that, for *every* $k$ Gaussian Mixture Model $\theta \in \Theta_{d,k}$, the negative log-likelihood $L(P \mid \theta)$ that $\theta$ generated the input set is provably approximately the same as the negative log-likelihood

$$L\big((C, w), \theta\big) := -\sum_{p \in C} w(p) \ln \sum_{i=1}^{k} \omega_i \frac{1}{\sqrt{\det(2\pi\Sigma_i)}} \exp(-\frac{1}{2}(p - \mu_i)^T \Sigma_i^{-1}(p - \mu_i)).$$

that $\theta$ generated $C$. More precisely, the approximation is up to a given multiplicative factor of $1 + \varepsilon$, i.e.,

$$L\big(P, \theta\big) \leqslant L\big((C, w), \theta\big) \leqslant (1 + \varepsilon)L\big(P, \theta\big). \quad (5)$$

The goal is to have a small coreset, i.e., to have a good trade off between $\varepsilon$ and the size of the coreset $C$. Running any (possibly inefficient) optimization algorithm to minimize equation 4 on the coreset, would then yield provable $(1 + \varepsilon)$-approximation to the original (full) data set $P$. In practice, we may run existing heuristics many times on the small coreset and take the best solution, in the hope to improve the result at the same running time. Combining such a coreset for in-memory data with existing techniques that we formalize in the appendix, enable support for streaming/dynamic distributed data (see Section F) "embarrassingly in parallel" Régin et al. (2013), in near-logarithmic time Bentley and Saxe (1980); Har-Peled and Mazumdar (2004); Indyk et al. (2014), and support hyper-parameter calibration Jubran et al. (2021); Maalouf et al. (2020); Mirzasoleiman et al. (2020).

**Projective clustering.** This paper forges a link between the family of $k$-GMM problems and the family of projective clustering problems, where the goal is to "cover" the points by strips or linear "slabs" of smallest width. Formally, the *fitting cost* $\text{dist}(P, S)$ of a given set $S$ of $k$ hyperplanes ($d - 1$ dimensional affine subspaces) to $P$ is the maximum over the $n$ distances between every point to its closest hyperplane. An optimal projective clustering $S^*$ is the set of $k$ hyperplanes that minimize this fitting cost, i.e., smallest width set that covers all the input points. Formally, by letting $H_{d,k}$ denote the union over every set of $k$-hyperplanes in $\mathbb{R}^d$,

$$\text{dist}(P, S^*) := \min_{S \in H_{d,k}} \text{dist}(P, S) := \min_{S \in H_{d,k}} \max_{p \in P} \text{dist}(p, S) := \min_{S \in H_{d,k}} \max_{p \in P} \min_{s \in S} \|p - s\|_2.$$

This is a generalization of the $k$-center problem or the Minimum Enclosing Ball (MEB) ($k = 1$), where the centers are hyperplanes instead of points.

## 2 RELATED WORK

The GMM fitting problem is one of the fundamental problems in machine learning which generalizes the notion of $k$-means clustering, where the covariance matrix that corresponds to each Gaussian is simply the identity matrix, i.e., its eigenvalues are all 1 and the Gaussian has the shape of a ball around some point $\mu \in \mathbb{R}^d$. It is also strongly related to Radial Basis Networks Alexandridis et al. (2017); Aljarah et al. (2018) and Radial Basis function Er et al. (2002); Pratiwi et al. (2015); Soundararajan et al. (2008); Wahba (1996). The problem is NP-hard when $k$ is part of the input Raghunathan et al. (2017) and many heuristics and approximation algorithms under different assumptions were suggested over the years; e.g. Gauvain and Lee (1994); Vlassis and Likas (2002); Zhang et al. (2003). The EM-algorithm is one of the common one in practice and is the one that is used in software libraries Gopi (2014); Hicks (2017); Kapoor et al. (2015); Townsend et al. (2016), as in our experimental results.

However, there are very little results that provably handle streaming (big) data, or insertions/deletions. The resulting coreset of this paper aims to change this situation, besides improving quality or running time, by enabling running existing algorithms on the coreset

**Coresets.** Coresets for many problems, especially in machine learning, were suggested over the recent years; see references in Feldman (2020). However, for $k$-GMMs we are aware of only one coreset Feldman et al. (2011); Lucic et al. (2017). Its construction is very similar to coresets for $k$-mean queries (distances to $k$-points) and its main disadvantage is that it approximates only "semi-spherical" Gaussians, which means that every covariance matrix of every $k$-GMM $\theta \in \Theta_{d,k}$ is close to the identity matrix. More precisely, the eigenvalues are both upper and lower bounded by 1 and $1/\varepsilon^2$ (or $\varepsilon$ and $1/\varepsilon$), respectively.

**Projective clustering.** A coreset for projective clustering whose size is $(\log n)^{O(1)}/\varepsilon^d$ was suggested in Edwards and Varadarajan (2005) for the case that each coordinate in the input set has a finite precision (i.e., represented by constant number of bits, such as "float" or "double" in C), and $d, k$ are constants. The dependencies on $d$ and $k$, as the assumption on the bounded precision are unavoidable; see Agarwal et al. (2005) and more references therein. More recently, the result was generalized sum (and not maximum) of distances Varadarajan and Xiao (2012b;c), a long-standing open problem in computational geometry.

## 3 OUR CONTRIBUTION

The primary results of this contribution are threefold: algorithms, experiments, and generalizations of existing results.

**Algorithms.** we present the first coreset construction for arbitrarily large $k$-GMM (Gaussian Mixture Models), whose size is sub-linear in the size of the input. More precisely, the algorithm gets as input a set $P$ of $n$ points, each consists of $d$ coordinates in a finite precision, a fixed integer $k \geqslant 1$ and a constant approximation error $\varepsilon \in (0, 1)$. It outputs a pair $(C, w)$ which, with high probability, satisfies equation 5 *simultaneously* for every $k$-GMM $\theta \in \Theta_{d,k}$. Here, "high probability" means a probability of failure that is less than, say, $\delta := 1/n^{10}$, the size of the coreset is $|C| \in O(\log^{O(1)}(n)/\varepsilon^2)$, i.e., near-logarithmic in $n$, and its construction time is $O(n \cdot |C|) = n \log^{O(1)} n$, i.e., near linear in $n$; see Theorem G.6 for exact details and dependencies on the parameters.

**Experiments.** On the practical side, we implemented our coreset construction and its sub-routines as open code. Reproducible experimental results are presented in Section 7. The experiments demonstrate how this coreset outperforms uniform sampling and the existing coreset for semi-spherical Gaussians, even for small sample size. One of our goals was to show that the theoretical asymptotical complexity bounds are heavily pessimistic, probably due to the assumption of worst case (possibly very artificial) input $P$ over *any* (not just optimal) $k$-GMM in $\Theta_{d,k}$, ignoring any structure in the data, and including very non-tight analysis (inequalities) in the proofs.

**Generalizations.**

In order to use these "folklore" or specific results for our application, we had to formalize and prove them for the general case first. These results correspond to the purple nodes in Fig. 1, Section A for bounding the VC-dimension, Section B for constructing coresets via sensitivities, Section E that reduces $\|\infty\|$ to $\|\cdot\|_1$ over distances, and Section F for supporting streaming/distributed input.

# 4 NOVEL TECHNIQUE: REDUCTION TO PROJECTIVE CLUSTERING.

The purple blocks in Fig. 1 represent the novel parts of this work, while the red ones are generalization of existing work.

The main challenge in computing coresets for $k$-GMMs, and the main reason that our algorithms are significantly more convoluted than the algorithm for semi-spherical Gaussians in Feldman et al. (2011); Lucic et al. (2017), is the fact that we cannot use the triangle inequality: a $k$-GMM whose eiganvalues have large mutual ratio behaves in some sense like a line or a hyperplane, and not as a point-center: points that are very far from each other might be close to the same Gaussian in the $k$-GMM mixture.

To handle this issue, we use a reduction to projective clustering, i.e., minimize maximum distance from a set $H$ of hyperplanes; see the linear grids in Fig. 2, Lemmas 6.2 and 6.3, and Section G. However, even though coresets for projective clustering exist Edwards and Varadarajan (2005); Varadarajan and Xiao (2012a;c), and are arguably the most complicated type of coresets, the reduction is not easy: handling *sum* of *exponential* and ln functions over matrices is different from *maximum* over *Euclidean* distances to hyperplanes. Moreover, we had to use *multiplicatively weighted* hyperplanes.

To this end, in Definition 6.1 the notion of $k$-SMM $y$ (subspace mixture models) in the higher dimensional space $\mathbb{R}^{2d+1}$ is introduced, inspired by the kernel trick for distances Schölkopf (2000). This is also related to the tedious technical challenges that we had to deal with in e.g. Lemma 6.3 via case analysis: we observed that there is a specific threshold ($e^{-\gamma r}$ there) such that if the weight $\omega_i$ of one of the Gaussians in $\theta$ is larger than this threshold, then we can adopt our coreset for $k$-SMMs. Otherwise, we can ignore this Gaussian and can use, by induction on $k$, the fact that the coreset hold for $(k-1)$-SMMs.

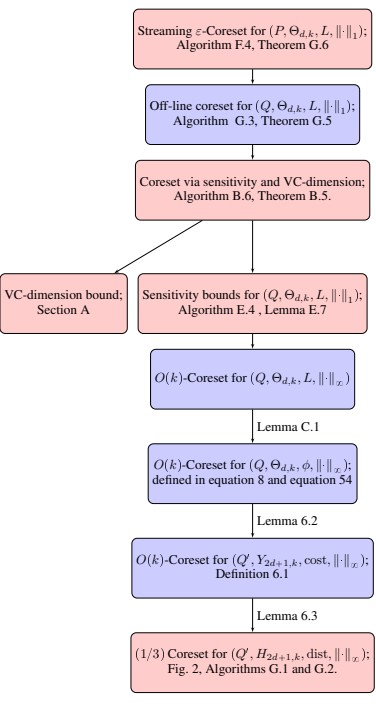

**Figure 1:**
Road-map of techniques & reductions.

# 5 ALGORITHM OVERVIEW

In this section, an $\varepsilon$-coreset $C$ for $(P, A, f, \|\cdot\|_\infty)$ implies that $C$ approximates $\max_{p \in P} f(p, a)$ for every $a \in A$, up to a factor of $\varepsilon$. Similarly, $\|\cdot\|_1$ instead if $\|\cdot\|_{\inf}$ implies $\sum_{p \in P} f(p, a)$. See formal Definition B.2.

The negative log-likelihood $L(P, \theta)$ of a $k$-GMM $\theta$ to a set $P$ of $n$ points in a finite precision is defined in equation 3. The generic streaming Algorithm F.4 computes the desired $\varepsilon$-coreset $(C, w)$ for $(P, \Theta_{d,k}, L, \|\cdot\|_1)$ whose size and update time per input point insertion is near-logarithmic in $n$, so total of near-linear time for all the points in $P$; see Theorem G.6. This is by $O(n)$ calls to our slower (quadratic time in $n$) Algorithm G.3 that computes an $\varepsilon$-coreset for $(Q, \Theta_{d,k}, L, \|\cdot\|_1)$ of small logarithmic-sized subsets $Q$ of $P$; see Theorem G.5. The technique is called *merge-reduce tree* Bentley and Saxe (1980); Har-Peled and Mazumdar (2004); Indyk et al. (2014) and is formalized in Section F.

Algorithm G.3 computes the importance of each point in $Q$, and then uses importance sampling, known as the Feldman-Langberg framework Feldman and Langberg (2011a), in Algorithm B.6, to sample the desired coreset for $(Q, \Theta_{d,k}, L, \|\cdot\|_1)$; see Theorem B.5. We formalized and generalized this framework in Section B. The importance for our specific case of $k$-GMMs is computed in

Algorithm E.4 as proved in Lemma E.7 via $O(|Q|)$ calls (iterations) to an $O(k)$-coreset construction for $(Q, \Theta_{d,k}, L, \|\cdot\|_\infty)$ that is removed from $Q$ in each iteration, until $Q$ is empty. The sensitivity (importance) of an (unweighted) point that was removed in the $i$th iteration is defined to be approximately $1/i$. This reduction from $\|\cdot\|$ to $\|\cdot\|_\infty$ is based on a general framework that we formalize in Section E, based on algorithms from modern computational geometry Agarwal et al. (2008); Varadarajan and Xiao (2012a). Since the sensitivities are computed for input coresets, we generalize Algorithm E.2 for weighted input set in Algorithm E.4.

It is left to compute an $O(k)$-coreset for $(Q, \Theta_{d,k}, L, \|\cdot\|_\infty)$. Lemma C.1 proves that it is an $O(k)$-coreset for $(Q, \Theta_{d,k}, \phi, \|\cdot\|_\infty)$, where $\phi$ is defined in equation 8 and equation 54, as a scaled version of $L$, similarly to Feldman et al. (2011); Lucic et al. (2017). In Definition 6.1, we introduce the union $Y_{d,k}$ over every $k$-SMM, which is a combination between Gaussian and subspaces in some sense, and its corresponding cost function. Lemma 6.2 and its proof in Section D implies that an $O(k)$-coreset for $(Q', Y_{2d+1,k}, \mathrm{cost}, \|\cdot\|_\infty)$ is the desired coreset for $(Q, \Theta_{d,k}, \phi, \|\cdot\|_\infty)$, where $Q'$ is the lifting of every point $Q \subseteq \mathbb{R}^d$ to $\mathbb{R}^{2d+1}$ by zero padding. By letting $H_{2d+1,k}$ denote the union over every set of $k$-hyperplanes in $\mathbb{R}^{2d+1}$, in Lemma 6.3 we prove that a $(1/3)$-coreset for $(Q', H_{2d+1,k}, \mathrm{dist}, \|\cdot\|_\infty)$ is the desired $O(k)$-coreset for $(Q', Y_{2d+1,\mathrm{cost},k}, \|\cdot\|_\infty)$.

Finally, it is left to compute an $O(1/3)$-coreset for $(Q', H_{2d+1,k}, \mathrm{dist}, \|\cdot\|_\infty)$, also known as projective clustering. This is the goal of Algorithm G.1 and its sub-routine Algorithm G.2, which is summarized in Fig. 2.

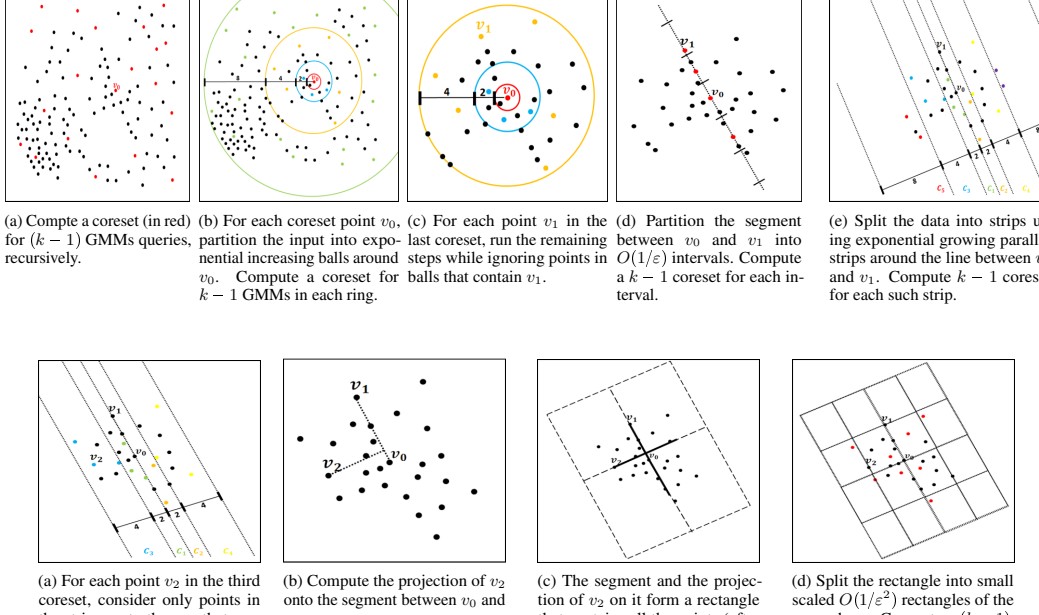

(a) Compte a coreset (in red) for $(k-1)$ GMMs queries, recursively.

(b) For each coreset point $v_0$, partition the input into exponential increasing balls around $v_0$. Compute a coreset for $k-1$ GMMs in each ring.

(c) For each point $v_1$ in the last coreset, run the remaining steps while ignoring points in balls that contain $v_1$.

(d) Partition the segment between $v_0$ and $v_1$ into $O(1/\varepsilon)$ intervals. Compute a $k-1$ coreset for each interval.

(e) Split the data into strips using exponential growing parallel strips around the line between $v_0$ and $v_1$. Compute $k-1$ coreset for each such strip.

(a) For each point $v_2$ in the third coreset, consider only points in the strips up to the one that contains $v_2$.

(b) Compute the projection of $v_2$ onto the segment between $v_0$ and $v_1$.

(c) The segment and the projection of $v_2$ on it form a rectangle that contains all the points (after scaling by a factor of 2).

(d) Split the rectangle into small scaled $O(1/\varepsilon^2)$ rectangles of the same shape. Compute a $(k-1)$-coreset for each rectangle.

**Figure 2**: Visualization of Algorithm G.1 and its sub-routine Algorithm G.2 in the appendix.

## 6 MAIN REDUCTIONS

This section presents the main reductions from the family of $k$-GMM from machine learning to $k$-SMM and then for hyperplanes in the context of projective clustering.

**Definition 6.1** ($k$-SMM). *The Euclidean distance between a point $p \in \mathbb{R}^d$ and a set $S \subseteq \mathbb{R}^d$ is denoted by*

$$\mathrm{dist}(p, S) = \inf_{s \in S} \|p - s\|_2.$$

*Its squared is denoted by $\mathrm{dist}^2(p, S) = \big(\mathrm{dist}(p, S)\big)^2$. Let $k \geqslant 1$ be an integer. A* subspace $k$-mixture model *in $\mathbb{R}^d$, or $k$-SMM for short, is a tuple $y = (W, \omega_1, \cdots, \omega_k, S_1, \cdots, S_k)$ where $W \geqslant 1$, $\omega$ is a*

*distribution vector, and each $S_i$ is an affine linear subspace of $\mathbb{R}^d$, for every $i \in [k]$. We also define $S(y) = \bigcup_{i=1}^{k} S_i$. The squared distance from a point $p \in \mathbb{R}^d$ to the k-SMM $y$ is defined by*

$$\text{cost}(p, y) := -\ln \sum_{i=1}^{k} \omega_i \exp(-W \text{dist}^2(p, S_i)). \tag{6}$$

*Similarly, $\text{cost}(P, y) := \sum_{p \in P} \text{cost}(p, y)$ for a finite set $P \subseteq \mathbb{R}^d$. The set $Y_{d,k}$ is the union over every k-SMM $y$ in $\mathbb{R}^d$ that satisfies $W \text{dist}_\infty(P, S(y)) \geq 1$.*

The following lemma enables us to compute coresets for $k$-SMMs instead of $k$-GMMs, while assuring a lower bound on the distance between the farthest point from the subspace.

**Lemma 6.2** (From $k$-GMM to $k$-SMM)**.** *Let $\theta \in \Theta_{d,k}$ be a $k$-GMM in $\mathbb{R}^d$. Then there is a k-SMM $y := (W, \omega_1', \cdots, \omega_k', S_1, \cdots, S_k) \in Y_{2d+1,k}$ that satisfies, for every $p \in \mathbb{R}^d$ and $p' := (p; 0; \cdots; 0) \in \mathbb{R}^{2d+1}$,*

$$W \text{dist}^2(p', S(y)) \geq 1, \tag{7}$$

*and*

$$\phi(p, \theta) := -\ln \sum_{i=1}^{k} \omega_i' \exp\left(-\frac{1}{2}(p - \mu_i)^T \Sigma_i^{-1}(p - \mu_i) - 1\right) = \text{cost}(p', y). \tag{8}$$

For every $k$-SMM $y$ , let

$$\text{cost}_\infty(P, y) = \max_{p \in P} \text{cost}(p, y), \tag{9}$$

be the point in $P$ with the maximum cost to $y$. For a set $P \subseteq \mathbb{R}^d$ of points, and a union $S = S_1 \cup \cdots \cup S_k$ of $k$ subspaces in $\mathbb{R}^d$, we define

$$\text{dist}_\infty(P, S) = \max_{p \in P} \text{dist}(p, S) \tag{10}$$

to be the distance of the farthest point in $P$ from $S$.    Recall that $H_{d,k}$ is the union over every set of $k$-hyperplanes in $\mathbb{R}^d$.

**Lemma 6.3** (From $k$-SMMs to hyperplanes )**.** *Let $P$ be a finite set of points in $\mathbb{R}^d$, and $k \geq 1$ be an integer. Then a $(1/3)$-coreset for $(P, H_{d,k}, \text{dist}, \|\cdot\|_\infty)$ is an $O(k)$-coreset for $(P, Y_{d,k}, \text{cost}, \|\cdot\|_\infty)$.*

*Proof.* Let $\psi := (1 + 1/3)^2$. Suppose that $C \subseteq P$ is a $(1/3)$-coreset of $(P, H_{d,k}, \text{dist}, \|\cdot\|_\infty)$. In particular, for every $k$-SMM $y \in Y_{d,k}$,

$$\text{dist}_\infty^2(P, S(y)) \leq (1 + 1/3)^2 \text{dist}_\infty^2(C, S(y)) = \psi \text{dist}_\infty^2(C, S(y)). \tag{11}$$

Let $\xi := 1$, and $h(k) := (4k)/\xi^2 + \psi + 1$. We will prove that every $k$-SMM $y \in Y_{d,k}$ satisfies

$$\text{cost}_\infty(P, y) \leq h(k)\text{cost}_\infty(C, y), \tag{12}$$

which will prove the lemma, since $h(k) \in O(k)$.

Indeed, let $y := (W, \omega_1, \ldots, \omega_k, S_1, \ldots, S_k) \in Y_{d,k}$, and $p^* \in \arg\max_{p \in P} \text{cost}(p, y)$. Without loss of generality, assume that

$$\text{dist}(p^*, S_1) = \text{dist}(p^*, S(y)). \tag{13}$$

That is, $S_1$ is a closest subspace to $p^*$ in $S(y)$. We first upper bound $r := \text{cost}_\infty(C, y)$ by

$$r = \text{cost}_\infty(C, y) \geq W \text{dist}_\infty^2(C, S(y)) \geq \frac{W \text{dist}_\infty^2(P, S(y))}{\psi} \geq \frac{\xi}{\psi}, \tag{14}$$

where the first inequality is by the definition of $S(y)$ (See Lemma D.2), the second inequality is by equation 11, and the third is by the definition of $Y_{d,k}$. The rest of the proof of equation 12 is by induction on $k$.

**Proof of equation 12 for the base case $k = 1$.** In this case, $y = (W, 1, S_1) \in Y_{d,1}$, and for every $p \in P$ we have

$$\text{cost}(p, y) = -\ln\left(1 \exp(-W \text{dist}^2(p, S_1))\right) = W \text{dist}^2(p, S_1) = W \text{dist}^2(p, S(y)), \tag{15}$$

where the first equality holds by equation 6. Hence, equation 12 follows as

$$\mathrm{cost}_\infty(P,y) = \mathrm{cost}(p^*,y) = \max_{p\in P}\mathrm{cost}(p,y) = \max_{p\in P}W\mathrm{dist}^2(p,S(y)) \tag{16}$$

$$= W\mathrm{dist}_\infty^2(P,S(y)) \leqslant W\psi\cdot\mathrm{dist}_\infty^2(C,S(y)) \leqslant \psi r \leqslant h(1)\mathrm{cost}_\infty(C,y), \tag{17}$$

where the equations in equation 16 hold by the definition of $p^*$ and equation 15, the first inequality in equation 17 is by equation 11, and the second inequality of equation 17 holds since $W\mathrm{dist}_\infty^2(C,S(y)) \leqslant r$ by equation 14. This proves equation 12 for the case $k=1$, as desired.

**Proof for the case $k \geqslant 2$.** Inductively assume that equation 12 holds for $k-1$. That is, for every $(k-1)$-SMM $y' \in Y_{d,k-1}$,

$$\mathrm{cost}_\infty(P,y') \leqslant h(k-1)\mathrm{cost}_\infty(C,y'). \tag{18}$$

We will use the bound

$$W\mathrm{dist}_\infty^2(C,S(y)) = W\max_{c\in C}\mathrm{dist}^2(c,S(y)) = \max_{c\in C}\left(-\ln\sum_{i=1}^k \omega_i\exp(-W\mathrm{dist}^2(c,S(y)))\right) \tag{19}$$

$$\leqslant \max_{c\in C}\left(-\ln\sum_{i=1}^k \omega_i\exp(-W\mathrm{dist}^2(c,S_i))\right) = \mathrm{cost}_\infty(C,y) = r, \tag{20}$$

where equation 19 holds since $S(y)$ is independent of $i$, and equation 20 holds since $\mathrm{dist}(c,S(y)) \leqslant \mathrm{dist}(c,S_i)$ by definition, for every $i \in [k]$. Therefore,

$$W\mathrm{dist}^2(p^*,S_1) = W\mathrm{dist}^2(p^*,S(y)) \leqslant W\mathrm{dist}_\infty^2(P,S(y)) \leqslant W\psi\mathrm{dist}_\infty^2(C,S(y)) \leqslant \psi r, \tag{21}$$

where the derivations are, respectively, by equation 13, equation 10, equation 11, and equation 19. Let $\alpha > 0$ be a parameter that will be defined later, and

$$\gamma := 1 - \frac{\ln(1-e^{-\alpha r})}{r}, \tag{22}$$

which is well defined since $r > 0$ by equation 14. The rest of the proof is by the following case analysis on the weight $\omega_1$ of the closest subspace $S_1 \in S(y)$ to $p^*$: Case **(i)** $\omega_1 \in [e^{-\gamma r}, 1]$, and Case **(ii)** $\omega_1 \in (0, e^{-\gamma r})$.

**Case (i):** $\omega_1 \in [e^{-\gamma r}, 1]$. In this case,

$$\mathrm{cost}_\infty(P,y) = \mathrm{cost}(p^*,y) = -\ln\left(\sum_{i=1}^k \omega_i\exp(-W\mathrm{dist}^2(p^*,S_i))\right) \tag{23}$$

$$\leqslant -\ln\left(\omega_1\exp(-W\mathrm{dist}^2(p^*,S_1))\right) \leqslant -\ln\left(\omega_1\exp(-\psi r)\right) \tag{24}$$

$$= -\ln\left(\exp(-\psi r + \ln\omega_1)\right) = \psi r - \ln\omega_1 \leqslant \psi r + \gamma r = (\psi + \gamma)\mathrm{cost}_\infty(C,y). \tag{25}$$

where equation 23 is by the definition of $\mathrm{cost}(p^*,y)$, the second inequality of equation 24 holds by equation 21, and the inequality of equation 25 holds by the assumption of Case (i).

**Case (ii):** $\omega_1 \in (0, e^{-\gamma r})$. We prove that in this case $\omega_1$ can be replaced by $\omega_1 = 0$, so that $y$ can be replaced by $y' \in Y_{d,k-1}$. We have $\omega_1 < e^{-\gamma r} \leqslant 1$, where the first inequality is by the assumption of Case (ii), and the second holds since $\gamma, r \geqslant 0$. Hence, $\beta := 1/(1-\omega_1)$ is well defined. Since $\sum_{i=1}^k \omega_i = 1$,

$$\sum_{i=2}^k \beta\omega_i = \frac{1}{(1-\omega_1)}\sum_{i=2}^k \omega_i = 1,$$

so the tuple $y' := (W, \beta\omega_2, \ldots, \beta\omega_k, S_2, \ldots, S_k)$ is a $(k-1)$-SMM. For every vector $z = (z_1, \cdots, z_k) \in [0,\infty)^k$ whose maximum is $\|z\|_\infty = z_1$, we have

$$\sum_{i=1}^k \omega_i z_i = \sum_{i=2}^k \beta\omega_i z_i(1-\omega_1) + \omega_1 z_1 = \sum_{i=2}^k \beta\omega_i z_i + \omega_1\left(z_1 - \sum_{i=2}^k \beta\omega_i z_i\right)$$

$$\geqslant \sum_{i=2}^k \beta\omega_i z_i + \omega_1(z_1 - \|z\|_\infty) = \sum_{i=2}^k \beta\omega_i z_i, \tag{26}$$

where the first equality holds since $\beta = 1/(1 - \omega_1)$, the inequality in equation 26 holds since $\beta(\omega_2, \cdots, \omega_k)$ is a distribution over $z \in (0, \infty)^k$, and the last equality holds by the definition of $z_1$. Taking the $-\ln$ of both sides in equation 26 yields

$$-\ln \sum_{i=1}^{k} \omega_i z_i \leqslant -\ln \sum_{i=2}^{k} \beta \omega_i z_i. \tag{27}$$

We can now obtain the bound

$$\text{cost}_\infty(P, y) = \text{cost}_\infty(p^*, y) = -\ln \sum_{i=1}^{k} \omega_i \exp(-W\text{dist}^2(p^*, S_i)) \tag{28}$$

$$\leqslant -\ln \sum_{i=2}^{k} \beta \omega_i \exp(-W\text{dist}^2(p^*, S_i)) \tag{29}$$

$$= \text{cost}(p^*, y') \leqslant \text{cost}_\infty(P, y') \leqslant h(k-1)\text{cost}_\infty(C, y'), \tag{30}$$

where equation 28 is by the definition of $\text{cost}(p^*, y)$, equation 29 holds by substituting $z_i := \exp(-W\text{dist}^2(p^*, S_i))$ for every $i \in [k]$ in equation 27, the first inequality of equation 30 is by the inductive assumption equation 18. It is left to bound $\text{cost}_\infty(C, y')$ by $\text{cost}_\infty(C, y)$.

Let $c^* \in \arg\max_{c \in C} \text{cost}(c, y')$. Hence,

$$e^{-r} \leqslant e^{-\text{cost}(c^*, y)} = \sum_{i=1}^{k} \omega_i \exp(-W\text{dist}^2(c^*, S_i)) \leqslant \omega_1 + \sum_{i=2}^{k} \omega_i \exp(-W\text{dist}^2(c^*, S_i)), \tag{31}$$

where the first inequality holds since $r = \text{cost}_\infty(C, y) \geqslant \text{cost}(c^*, y)$, and the second inequality holds since $W\text{dist}^2(c^*, S_k) \geqslant 0$. Subtracting $\omega_1$ from both sides of equation 31 and multiplying by $\beta$ yields

$$\beta(e^{-r} - \omega_1) \leqslant \sum_{i=2}^{k} \beta \omega_i \exp(-W\text{dist}^2(c^*, S_i)) = e^{-\text{cost}(c^*, y')} = e^{-\text{cost}_\infty(C, y')}. \tag{32}$$

Therefore,

$$\text{cost}_\infty(C, y') \leqslant -\ln(\beta(e^{-r} - \omega_1)) \leqslant -\ln(e^{-r} - e^{-\gamma r}) \tag{33}$$

$$= -\ln(e^{-r} - e^{-r + \ln(1 - \exp\{-\alpha r\})}) = -\ln\left(e^{-r}(1 - (1 - e^{-\alpha r}))\right) \tag{34}$$

$$= -\ln(\exp(-r - \alpha r)) = r(1 + \alpha) = (1 + \alpha)\text{cost}_\infty(C, y), \tag{35}$$

where the inequalities in equation 33 hold by taking the $\ln$ of both sides of equation 32 and multiplying by $(-1)$, and since $\beta > 1$, respectively. The left equality of equation 34 is by definition equation 22 of $\gamma$, and the last equality is by the definition of $r$. Plugging equation 33 in equation 35 yields a bound of $\text{cost}_\infty(P, y)$ with respect to $\text{cost}_\infty(C, y)$

$$\text{cost}_\infty(P, y) \leqslant h(k-1)\text{cost}_\infty(C, y') \leqslant (1 + \alpha)h(k-1)\text{cost}_\infty(C, y). \tag{36}$$

**Calibrating $\alpha$.** By equation 25 and equation 36, in order to prove equation 11 it is left to bound $\psi + \gamma$ and $(1 + \alpha)h(k-1)$, respectively, by $h(k) = (4k)/\xi^2 + \psi + 1$. Let $\alpha := 1/k$. Hence,

$$(1 + \alpha)h(k-1) \leqslant \frac{k}{\xi^2}(1 + \alpha) = \frac{k}{\xi^2}\left(1 + \frac{1}{k}\right) = \frac{k}{\xi^2} + \frac{1}{\xi^2} = \frac{k+1}{\xi^2} \leqslant h(k). \tag{37}$$

To bound $\gamma$, note that $(t-1)/t \leqslant \ln t \leqslant t - 1$ for every $t > 0$. We thus have, for every $x > 0$,

$$-\ln(1 - e^{-x}) \leqslant \frac{e^{-x}}{1 - e^{-x}} = \frac{1}{e^x - 1} \leqslant \frac{1}{x}, \tag{38}$$

by substituting $t = 1 - e^{-x}$ in the first inequality, and $t = e^{-x}$ in the second inequality. Therefore,

$$\psi + \gamma = \psi + 1 - \frac{\ln(1 - e^{-r/k})}{r} \leqslant \psi + 1 + \frac{k/r}{r} = \psi + 1 + \frac{k}{r^2} \leqslant \psi + 1 + \frac{4k}{\xi^2} = h(k), \tag{39}$$

where the inequality holds since by substituting $x := -r/k$ in equation 38. Plugging equation 37 and equation 39 in equation 36 and equation 25, respectively, proves equation 12 for $k \geqslant 2$. $\qquad\square$

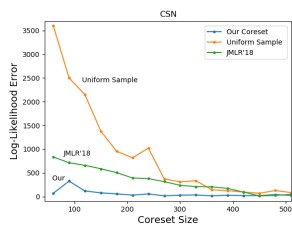 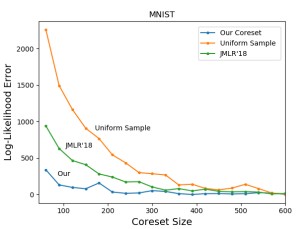 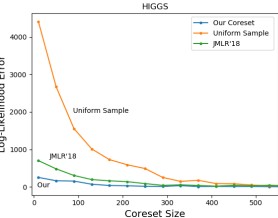

**Figure 3**: Results for three reconstructed experiments on three real datasets, two from Feldman et al. (2011) and a third from Lucic et al. (2017). For every fixed sample size, our coresets enjoy smaller approximation errors.

## 7 EXPERIMENTAL RESULTS

We implemented our main coreset construction and its subroutines into an open sourced Python's library removed due to anonymization (2025). Our experiments were applied on three public real-world datasets from Feldman et al. (2011); Lucic et al. (2017); CSN cell phone accelerometer data contains $n = 40,000$ feature vectors in $d = 17$ dimensions, MNIST handwritten digits contains $n = 60,000$ grayscale 784-pixel (28x28) images of handwritten digits which were projected to $d = 100$ feature vectors, and Higgs high-energy physics dataset which contains $n = 11,000,000$ instances in $d = 150$ dimensions describing signal processes which produce Higgs bosons and background processes which do not Baldi et al. (2014). See appendix and Feldman et al. (2011); Lucic et al. (2017) for more details.

We trained a $k$-GMM $G_{\text{org}}$ using each dataset $P$ with a common python library (Pomegranate), with $k = 6$, $k = 10$, and $k = 15$ GMMs, respectively, for the CSN, MNIST, and Higgs datasets, and then computed the negative log-likelihood $\ell(G_{\text{org}}, P)$. We used the entire data set several times to train a target GMM $G_{\text{trg}}$, and computed the average log-likelihood $\ell(G_{\text{trg}}, P)$ of $P$ using $G_{\text{trg}}$. Finally, we defined the optimal log-likelihood as $\ell_{\text{opt}} := |\ell(G_{\text{org}}, T) - \ell(G_{\text{trg}}, T)|$. We then compare the results on uniformly sampled subsets of sizes between 20 and 5000, using $k = 10$ and fit GMMs using EM. Unlike the theoretical and very pessimistic worse-case bounds (for all these types of samples), we use the algorithms only for compute the distribution on the input points in $P$. Then we sample the desired number of points, independently of the actual bounds and their dependency on $d$, $k$, and $\varepsilon$.

The coreset's construction time was similar to Lucic et al. (2017) for every dataset and coreset size. For a coreset of size 20 (first point in **3** ), the approximation error using CSN dataset was 10 times smaller than Lucic et al. (2017) and 40 times smaller than uniform sampling, and using MNIST Lucic et al. (2017) coreset approximation error was twice than our approximated error. For some coreset sizes on Higgs dataset, our approximation error was up to 3 times smaller than Lucic et al. (2017).

## 8 CONCLUSION AND OPEN PROBLEMS

We provided algorithms that compute, with high probability, an $\varepsilon$-coresets for the family of mixtures of $k$ Gaussian models. The coreset can be maintained for streaming, distributed and dynamic input data using existing techniques. The key idea is a serious of reductions from likelihood of Gaussians to maximum distance to $k$ subspaces, known as the projective clustering problem.

Due to the applications of projective clustering for many other types of measures and shapes via "linearization" Agarwal et al. (2004); Har-Peled (2006); Marom and Feldman (2019), we expect that the reduction in this paper would open the door for many other related techniques such as kernels Scholkopf and Smola (2018), Deep Gaussian Learning Viroli and McLachlan (2019), Deep Autoencoding Gaussian Mixture Model Zong et al. (2018), Gaussian Mixture Convolution Networks Celarek et al. (2022), or in computer vision for registration Yuan et al. (2020) or Gaussian Splatting, where the input (and not output) is a set of Gaussians Wu et al. (2024).

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

# A  VC-DIMENSION BOUND

Coreset construction for a tuple $(P, Y, f, \mathrm{loss})$ using the Feldman-Langberg framework, is based on the following general result that bounds the VC-dimension of a set via the time it takes to answer a query, i.e, the time to compute $f(y)$ for a specific $y \in Y$.

**Definition A.1** (basic operations). *Let $P$ and $Y$ be two sets, and $f : P \times Y \to \mathbb{R}$ be a function that can be evaluated by an algorithm that gets $(p, y) \in P \times Y$ and returns $f(p, y)$ after no more than $z$ of the following operations:*

1. *the exponential function $\alpha \mapsto e^{\alpha}$ on real numbers,*

2. *the arithmetic operations $+, -, \times,$ and $/$ on real numbers,*

3. *jumps conditioned on $>, \geqslant, <, \leqslant, =,$ and $\neq$ comparisons of real numbers.*

*If the $z$ operations include no more than $k$ in which the exponential function is evaluated, then we say that the function $f$ can be evaluated using $z$ operations that include $k$ exponential operations.*

The dimension of a coreset for a tuple $(P, Y, f, \mathrm{loss})$ in the Feldman-Langberg framework depends on and the dimension of the first three items $(P, Y, f)$, and is strongly related to VC-dimension; see Definition B.3. The following is a variant of (Anthony and Bartlett, 2009, Theorem 8.14) that bounds the VC-dimension of a pair $(P, \mathrm{ranges})$, adapted for our version of dimension (in particular, we exchange $d$ and $m$ in (Anthony and Bartlett, 2009, Theorem 8.14)). It provides a not-so-tight but very generic upper bound for the case where $P \subseteq \mathbb{R}^d$, $Y' \subseteq \mathbb{R}^m$, and $f(p, y')$ can be computed via small (less than $|P|$) basic operations, as proved in Corollary A.3.

**Theorem A.2** (Variant of (Anthony and Bartlett, 2009, Theorem 8.14)). *Let $h : \mathbb{R}^d \times \mathbb{R}^m \to \{0, 1\}$ be a binary function that can be evaluated using $O(z)$ operations that include $O(k)$ exponential operations; see Definition A.1. Then $\dim(\mathbb{R}^d, \mathbb{R}^m, h) \in O(m^2k^2 + mkz)$; see Definition B.3.*

**Corollary A.3.** *Let $P \subseteq \mathbb{R}^d$, $Y' \subseteq \mathbb{R}^m$ and $f : P \times Y' \to [0, \infty)$. If, for every $p \in P$ and $y' \in Y'$ the value $f(p, y')$ can be computed in $O(z)$ arithmetic operations that include $O(k)$ exponential operations, then $\dim(P, Y', f) \in O(m^2k^2 + mkz)$.*

*Proof.* It suffices to prove the desired bound on the dimension of $(\mathbb{R}^d, \mathbb{R}^m, f)$, i.e. assume that $Y' = \mathbb{R}^m$, since the VC-dimension, as the dimension of $(P, Y, f)$, is monotonic in the cardinality of the set $Y'$ of queries and set $P$. Suppose that $S$ is the largest subset $S \subseteq \mathbb{R}^d$ such that

$$\left| \left\{ \mathrm{range}_{S,f}(y, r) \mid r \in \mathbb{R}, y \in \mathbb{R}^m \right\} \right| = \{ p \in S \mid f(p, y) \leqslant r \} = 2^{|S|}; \tag{40}$$

see Definition B.3. We need to upper bound the size of $S$.

Define $h : \mathbb{R}^d \times \mathbb{R}^{m+1} \to \{0, 1\}$ such that $h(p, x) = 0$ if and only if there is $y' \in \mathbb{R}^m$ and $r \in \mathbb{R}$ such that $x = (y'; r)$ and $f(p, y') \leqslant r$. We then have, for every such $x := (y'; r)$ in $\mathbb{R}^{m+1}$,

$$\mathrm{range}_{S,h}(x, 0) = \mathrm{range}_{S,h}((y'; r), 0) = \left\{ p \in S \mid h(p, (y'; r)) \leqslant 0 \right\}$$
$$= \left\{ p \in S \mid h(p, (y'; r)) = 0 \right\} = \left\{ p \in S \mid f(p, y') \leqslant r \right\} = \mathrm{range}_{S,f}(y', r).$$

Hence,

$$\left| \left\{ \mathrm{range}_{S,h}(x, r') \mid x \in \mathbb{R}^{m+1}, r' \in \mathbb{R} \right\} \right| \geqslant \left| \left\{ \mathrm{range}_{S,h}(x, 0) \mid x \in \mathbb{R}^{m+1} \right\} \right|$$
$$= \left| \left\{ \mathrm{range}_{S,f}(y', r) \mid r \in \mathbb{R}, y' \in \mathbb{R}^m \right\} \right| = 2^{|S|}, \tag{41}$$

where the last equality is by equation 40. Since $\mathrm{range}_{S,h}(x, r')$ is a subset of $S$, and there are at most $2^{|S|}$ such subsets, equation 41 implies

$$\left| \left\{ \mathrm{range}_{S,h}(x, r') \mid x \in \mathbb{R}^{m+1}, r' \in \mathbb{R} \right\} \right| = 2^{|S|}. \tag{42}$$

By Theorem A.2, the dimension of $(\mathbb{R}^d, \mathbb{R}^{m+1}, h)$ is $d' \in O(m^2k^2 + mkz)$. Hence, the size of the largest subset $S \subseteq \mathbb{R}^d$ that satisfies equation 42 is $|S| \leqslant d'$. By equation 40 and Definition B.3, $d'$ also an upper bound for the dimension of $\dim(P, Y', f) \leqslant d' \in O(m^2k^2 + mkz)$. $\qquad \square$

The following corollary generalizes Theorem 12 in Lucic et al. (2017) from $\mathrm{cost}$ to $f$, and from semi-spherical Gaussians instead of any Gaussian, using similar approach.

**Corollary A.4** (Generalization of (Lucic et al., 2017, Theorem 12))**.** *Let $k \geqslant 1$ be an integer and $(P, w)$ be a weighted set such that $P \subseteq \mathbb{R}^d$ is finite. Let $f : \mathbb{R}^d \times \Theta_{d,k} \to [0, \infty)$ be the function that maps every $p \in \mathbb{R}^d$ and $\theta \in \Theta_{d,k}$ to*

$$f(p, \theta) = \frac{\phi(p, \theta)}{\phi((P, w), \theta)}, \tag{43}$$

*if the denominator is positive, and $f(p, \theta) = 0$ otherwise. Then $\dim(P, \Theta_{d,k}, f) \in O(d^2 k^4)$.*

*Proof.* For every $p \in \mathbb{R}^d$, let $p' := (p; 0; \cdots ; 0) \in \mathbb{R}^{2d+1}$. Let $\xi = 1$ and $P, := \{p' \mid p \in P\}$. For every $k$-SMM $y = (W, \omega_1, \cdots, \omega_k, S_1, \cdots, S_k) \in Y_{2d+1,k}$ and $p, \in P'$, define (as in equation 6)

$$\mathrm{cost}(p', y) = -\ln \sum_{i=1}^{k} \omega_i \exp(-W \mathrm{dist}^2(p', S_i)),$$

and

$$g(p', y) = \frac{\mathrm{cost}(p', y)}{\sum_{q \in P} w(q) \mathrm{cost}(q', y)},$$

if the denominator is positive, and $g(p', y) = 0$ otherwise. We first prove that

$$\dim(P, \Theta_{d,k}, f) \leqslant \dim(P', Y_{2d+1,k}, g). \tag{44}$$

Indeed, let $S$ be the largest subset of $P$ that satisfies

$$\left| \{S \cap \mathrm{range} \mid \mathrm{range} \in \mathrm{ranges}(P, \Theta_{d,k}, f)\} \right| = 2^{|S|}, \tag{45}$$

where $\mathrm{ranges}(\cdot, \cdot, \cdot)$ is defined in equation 49. Hence,

$$\begin{aligned} 2^{|S|} &= \left| \{S \cap \mathrm{range} \mid \mathrm{range} \in \mathrm{ranges}(P, \Theta_{d,k}, f)\} \right| \\ &= \left| \{\mathrm{range}_{S,f}(\theta, r) \mid r \geqslant 0, \theta \in \Theta_{d,k}\} \right|. \end{aligned} \tag{46}$$

Let $S' = \{p' \mid p \in S\}$. Let $\theta \in \Theta_{d,k}$ be a $k$-GMM. By Lemma 6.2, there is a $k$-SMM $y \in Y_{2d+1,k}$ that satisfies $\phi(p, \theta) = \mathrm{cost}(p', y)$ for every $p \in \mathbb{R}^d$. Hence, for every $p \in S$, there is a corresponding point $p' \in S'$ that satisfies

$$f(p, \theta) = \frac{\phi(p, \theta)}{\sum_{q \in P} w(q) \phi(q, \theta)} = \frac{\mathrm{cost}(p', y)}{\sum_{q \in P} w(q) \mathrm{cost}(q', y)} = g(p', y).$$

Here we assumed $\phi(p, \theta) > 0$, otherwise $f(p, \theta) = 0 = g(p', y)$ still holds. In particular, for every $r \geqslant 0$, the set

$$\mathrm{range}_{S,f}(\theta, r) = \{p \in S \mid f(p, \theta) \leqslant r\}$$

has a corresponding distinct set

$$\mathrm{range}_{S',g}(y, r) = \left\{ p' \in S' \mid g(p', y) \leqslant r \right\}.$$

Therefore,

$$2^{|S|} = \left| \left\{ \mathrm{range}_{S,f}(\theta, r) \mid r \geqslant 0, \theta \in \Theta_{d,k} \right\} \right| \leqslant \left| \left\{ \mathrm{range}_{S',g}(y, r) \mid r \geqslant 0, y \in Y_{2d+1,k} \right\} \right|, \tag{47}$$

where the first equality is by equation 46. Since the last expression is a set of subsets from $S'$, its size is upper bounded by $2^{|S'|} = 2^{|S|}$. Together with equation 47 we obtain,

$$2^{|S'|} = \left| \left\{ \mathrm{range}_{S',g}(y, r) \mid r \geqslant 0, y \in Y_{2d+1,k} \right\} \right|,$$

so $|S'| \leqslant \dim(P', Y_{2d+1,k}, g)$ by the definition of $\dim(P', Y_{2d+1}, g)$. The last inequality proves equation 44 as

$$2^{\dim(P, \Theta_{d,k}, f)} = 2^{|S|} = 2^{|S'|} \leqslant 2^{\dim(P', Y_{2d+1,k}, g)},$$

where the first equality is by the definition of $S$ and $\dim(P, \Theta_{d,k}, f)$.

Next, we upper bound $\dim(P', Y_{2d+1,k}, g)$. Indeed, for fixed $r > 0$ and $r' := r \sum_{q \in P} w(q)\mathrm{cost}(q', y)$,

$$\{p \in S \mid g(p', y) \leqslant r\} = \left\{ p \in S \mid \frac{\mathrm{cost}(p', y)}{\sum_{q \in P} w(q)\mathrm{cost}(q', y)} \leqslant r \right\} = \{p \in S \mid \mathrm{cost}(p', y) \leqslant r'\},$$

and the equalities trivially holds for $r = 0$. Hence, it suffices to bound the dimension of $(P', Y_{2d+1,k}, \mathrm{cost})$ which has the same dimension if the function $\mathrm{cost}(\cdot, \cdot)$ is replaced by $\mathrm{cost}' = e^{\mathrm{cost}(\cdot, \cdot)}$, i.e.,

$$\dim(P', Y_{2d+1,k}, g) = \dim(P', Y_{2d+1,k}, \mathrm{cost}) = \dim(P', Y_{2d+1,k}, \mathrm{cost}'). \tag{48}$$

For every $y \in Y$, let $y' \in \mathbb{R}^m$ denote the concatenation of the parameters $W, \omega$ and the orthogonal unit vectors and translations of $S_1, \cdots, S_k$ into a single vector of length $m = O(dk)$. Let $\mathrm{cost}' : P' \times Y_{2d+1,k} \to \mathbb{R}$ be a function that maps every $(p, y) \in P \times Y_{2d+1,k}$ to

$$\mathrm{cost}'(p', y') = e^{\mathrm{cost}(p,y)} = -\sum_{i=1}^{k} \omega_i \exp(-W \mathrm{dist}^2(p, S_i)),$$

which can be computer via $z = O(m)$ operations that include $k + 1$ exponential operations; see Definition A.1. Applying Corollary A.3 yields that the dimension of $(P', Y_{2d+1,k}, \mathrm{cost}')$ is

$$\dim(P', Y, \mathrm{cost}') \in O(d^2 k^4 + dk^3) = O(d^2 k^4).$$

Combining this with equation 48 and equation 44 proves the corollary as

$$\dim(P, \Theta_{d,k}, f) \leqslant \dim(P', Y_{2d+1,k}, g) = \dim(P', Y_{2d+1,k}, \mathrm{cost}') \in O(d^2 k^4).$$

$\square$

As stated in Feldman et al. (2011); Lucic et al. (2017), the lower-bound of $\Omega(kd^2)$ was established by Akama and Irie (2011) for the dimension of $(P, Y, f)$ above. It is an open problem whether this gap can be closed further in the general setting.

## B  FROM CORESETS TO SENSITIVITY AND VC-DIMENSION

Our next goal is to compute coresets for $k$-SMMs, which would be used for $k$-GMMs as explained in the previous chapter. To this end, we introduce an existing generic framework for coreset construc­tions.

A coreset is problem dependent, and the problem is defined by four items: the input weighted set, the possible set of queries (models) that we want to approximate, the cost function per point, and the overall loss calculation. In this paper, the input is usually a set of points in $\mathbb{R}^d$ or $\mathbb{R}^{2d+1}$, but for the streaming case in Section F we compute coreset for union of (weighted) coresets and thus weights will be needed. The queries are either Gaussians or subspaces (for projective clustering), the cost/kernel $f$ is the $\phi$ function, negative log-likelihood $L$, or Euclidean distance, and the loss would be either maximum or sum over costs in Sections E and D respectively.

**Definition B.1** (query space). *Let $Y$ be a (possibly infinite) set called* query set, *$P' = (P, w)$ be a weighted set called the* input set, *$f : P \times Y \to [0, \infty)$ be called a* kernel or cost function, *and* loss *be a function that assigns a non-negative real number for every real vector. The tuple $(P', Y, f, \mathrm{loss})$ is called a* query space. *For every weighted set $C' = (C, u)$ such that $C = \{p_1, \cdots, p_m\} \subseteq P$, and every $y \in Y$, we define the overall fitting error of $C'$ to $y$ by*

$$f_{\mathrm{loss}}(C', y) := \mathrm{loss}((w(p)f(p, y))_{p \in C}) = \mathrm{loss}(w(p_1)f(p_1, y), \cdots, w(p_m)f(p_m, y)).$$

A coreset that approximates a set of models (queries) is defined as follows, where equation 5 is a special case. A weighted set is defined as s a pair that consists of a set $P$ and a (weight) function $w : P \to [0, \infty)$. We define $(1 \pm \varepsilon) := [1 - \varepsilon, 1 + \varepsilon]$ for $\varepsilon > 0$, and $aX = \{ax \mid x \in X\}$ for every set $X$ and $a \in \mathbb{R}$.

**Definition B.2** ($\varepsilon$-coreset). *Let $P' = (P, w)$ be a weighted set. For an approximation error $\varepsilon > 0$, the weighted set $C' = (C, u)$ is called an $\varepsilon$-coreset for a query space $(P', Y, f, \mathrm{loss})$, if $C \subseteq P$, and for every $y \in Y$ we have*

$$f_{\mathrm{loss}}(P', y) \in (1 \pm \varepsilon) f_{\mathrm{loss}}(C', y).$$

The dimension of a query space $(P, Y, f)$ is the VC-dimension of the range space that it induced, as defined below. The classic VC-dimension was defined for sets and subset and here we generalize it to query spaces, following Feldman and Langberg (2011b).

**Definition B.3** (Dimension for a query space Vapnik and Chervonenkis (1971); Braverman et al. (2016); Feldman and Langberg (2011b)). *For a set $P$ and a set ranges of subsets of $P$, the VC-dimension of $(P, \mathrm{ranges})$ is the size $|C|$ of the largest subset $C \subseteq P$ such that*

$$\left| \{ C \cap \mathrm{range} \mid \mathrm{range} \in \mathrm{ranges} \} \right| = 2^{|C|}.$$

*Let $Y$ be a set and $f : P \times Y \to \mathbb{R}$. For every query $y \in Y$, and $r \in \mathbb{R}$ we define the set*

$$\mathrm{range}_{P,f}(y, r) := \{ p \in P \mid f(p, y) \leqslant r \} .$$

*and*

$$\mathrm{ranges} := \mathrm{ranges}(P, Y, f) := \left\{ \mathrm{range}_{P,f}(y, r) \mid y \in Y, r \in \mathbb{R} \right\} . \tag{49}$$

*The dimension $\dim(P, Y, f)$ is defined as the VC-dimension of $(P, \mathrm{ranges})$.*

We use the general reduction for computing coresets for sum over the cost of each point, by bounding its sensitivity (importance) as defined below. The size of the coreset then depends near linearly on the sum of these bounds via Braverman et al. (2016) following the quadratic bound in Feldman and Langberg (2011b). The rest of the paper will be devoted mainly to compute such a bound.

**Definition B.4** (sensitivity). *Let $P' = (P, w)$ be a weighted set, and $((P, w), Y, \mathrm{cost}, \|\cdot\|_1)$ be a query space. The function $s^* : P \to [0, \infty)$ is the sensitivity of $(P', Y, \mathrm{cost})$ if*

$$s^*(p) = \sup_y \frac{w(p)\mathrm{cost}(p, y)}{\sum_{q \in P} w(p)\mathrm{cost}(q, y)},$$

*for every $p \in P$, where the sup is over every $y \in Y$ such that the denominator is non-zero.*

*The function $s : P \to [0, \infty)$ is a sensitivity bound for $(P', Y, \mathrm{cost})$ if $s(p) \geqslant s^*(p)$ for every $p \in P$. The total sensitivity of $s$ is defined as*

$$t = \sum_{p \in P} s(p).$$

The following theorem proves that a coreset can be computed by sampling according to sensitivity of points. The size of the coreset depends on the total sensitivity and the complexity (VC-dimension) of the query space, as well as the desired error $\varepsilon$ and probability $\delta$ of failure.

**Theorem B.5** (coreset from sensitivities Braverman et al. (2016); Feldman and Langberg (2011a)). *Let*

- $((P, w), Y, \mathrm{cost}, \|\cdot\|_1)$ *be a query space, and $n := |P|$.*

- $f : P \times Y \to [0, \infty)$ *such that for every $p \in P$ and $y \in Y$,*

$$f(p, y) = \begin{cases} \frac{w(p)\mathrm{cost}(p,y)}{\sum_{q \in P} w(q)\mathrm{cost}(q,y)} & \sum_{q \in P} w(q)\mathrm{cost}(q, y) > 0 \\ 0 & \sum_{q \in P} w(q)\mathrm{cost}(q, y) = 0, \end{cases}$$

- $s : P \to [0, \infty)$ *be a sensitivity bound of $((P, w), Y, \mathrm{cost})$, and $t = \sum_{p \in P} s(p)$ be its total sensitivity.*

- $\varepsilon, \delta \in (0, 1)$,

- $c > 0$ *be a universal constant that can be determined from the proof,*

- 

$$m \geqslant \frac{c(t + 1)}{\varepsilon^2} \left( \dim(P, Y, f) \log(t + 1) + \log\left(\frac{1}{\delta}\right) \right), and$$

- $(C, u)$ be the output weighted set of a call to $\text{CORESET}(P, w, s, m)$; see Algorithm B.6.

*Then $(i)$–$(v)$ hold as follows.*

    *(i) With probability at least $1 - \delta$, $C$ is an $\varepsilon$-coreset of $((P, w), Y, \text{cost}, \|\cdot\|_1)$.*

    *(ii) $|C| = m$.*

    *(iii) $(C, u)$ can be computed in $O(n)$ time, given $(P, w, s, m)$.*

    *(iv) $u(p) \in [w(p), \sum_{q \in P} w(q)/m]$ for every $p \in C$.*

    *(v) $\sum_{p \in P} w(p) = \sum_{q \in C} u(q)$.*

The last two properties would be required to support streaming in Section F, and keep the overall weight of the coreset, that depends on sum and maximum over input weight.

---

**Algorithm B.6:** $\text{CORESET}(P, w, s, m)$

---

**Input:**    A weighted set $(P, w)$, where $\sum_{p \in P} w(p) > 0$, $s : P \to [0, \infty)$, and an integer $m \geq 1$.
**Output:**  A weighted set $(C, u)$ that satisfies Theorem B.5.

```
// Add small importance to each point to bound coreset's
   weights
```
1  $s'(p) := s(p) + \dfrac{w(p)}{\sum_{q \in P} w(q)}$
```
// Add very important points to the coreset
```
2  $C := \left\{ p \in P \mid \frac{s'(p)}{\sum_{q \in P} s'(q)} \geq \frac{1}{m} \right\}$
3  **for** *every $p \in C$* **do**
4     $u'(p) := w(p)$
5  $Q := P \backslash C$
6  **for** *$m$ iterations* **do**
7     Sample a point $q$ from $Q$, where $p \in Q$ is chosen with probability $\Pr(p) := \dfrac{s'(p)}{\sum_{p' \in Q} s'(p')}$
8     $C := C \cup \{q\}$
9     $u'(q) := \frac{w(q)}{m \cdot \Pr(q)}$
10 **for** *every $p \in C$* **do**
```
   // sum of weights in the coreset and input set should be the
      same
```
11    $u(p) := u'(p) \cdot \dfrac{\sum_{p' \in P} w(p')}{\sum_{q \in C} u'(q)}$
12 **return** $(C, u)$

---

## C   From Likelihood to $\phi$ approximation

In this section we continue the discussion in Section 1. Recall that a $k$-GMM

$$\theta := \big\{ (\omega_i, \Sigma_i, \mu_i) \big\}_{i=1}^{k} := \big\{ (\omega_1, \mu_1, \Sigma_1), \cdots, (\omega_k, \mu_k, \Sigma_k) \big\}, \tag{50}$$

was defined in equation 1. Inspired by Feldman et al. (2011), let

$$Z(\theta) := \sum_{i=1}^{k} \frac{\omega_i e}{\sqrt{\det(2\pi \Sigma_i)}}, \tag{51}$$

and, for every $i \in [k]$, let

$$\omega_i' := \frac{\omega_i e}{Z(\theta) \sqrt{\det(2\pi \Sigma_i)}}. \tag{52}$$

Hereby $Z$ is a normalizer that asserts that $\omega'$ is a distribution vector. As was shown in Feldman et al. (2011), the negative log-likelihood of a point $p \in \mathbb{R}^d$ and a $k$-GMM $\theta \in \Theta_{d,k}$ can be decomposed into two parts, where one of them depends only on $\theta$, and not on $p$, as follows:

$$L(p, \theta) = \phi(p, \theta) - \ln Z(\theta), \tag{53}$$

where

$$\phi(p, \theta) := -\ln \sum_{i=1}^{k} \omega_i' \exp\left(-\frac{1}{2}(p - \mu_i)^T \Sigma_i^{-1}(p - \mu_i) - 1\right). \tag{54}$$

Letting $\phi(P, \theta) := \sum_{p \in P} \phi(p, \theta)$ and summing equation 54 over every $p \in P$ yields

$$L(P, \theta) = \sum_{p \in P} L(p, \theta) = \phi(P, \theta) - n \ln Z(\theta). \tag{55}$$

Since $L(P, \theta)$ might be negative, it does not make sense to consider coreset or multiplicative approximation for $L$. Instead, we approximate $\phi(P, \theta)$ which is always positive. If $\ln Z(\theta) < 0$, then $L(P, \theta) > \phi(\theta)$, so in this case we also have a coreset on approximation for $L(P, \theta)$. We consider the conditions for this case in the following paragraph and lemma.

For a weighted set $(C, u)$, we generalize the above definitions, by letting

$$\phi((C, w), \theta) := \sum_{c \in C} w(c)\phi(c, \theta) = \sum_{c \in C} -w(c)\ln \sum_{i=1}^{k} \omega_i' \exp\left(-\frac{1}{2}(c - \mu_i)^T \Sigma_i^{-1}(c - \mu_i) - 1\right),$$

and

$$L((C, w), \theta) := \sum_{c \in C} u(c)L(c, \theta) = \phi((C, w), \theta) - \sum_{c \in C} u(c)\ln Z(\theta). \tag{56}$$

The coreset for the $\phi$ cost function is similar to the log-likelihood as explained above. Nevertheless, for the case of $\phi$ we would have coresets for any $k$-GMM. Moreover, it would be easier to work with this function in the rest of the paper. The justification and reduction for the likelihood is explained in the next section.

As noted in (Feldman et al., 2011, Section 2), $Z(\theta)$ can be computed *exactly* and independently of the set $P$. Furthermore, the function $\phi$ captures all dependencies of $L(P, \theta)$ on $\theta$.

Moreover, if either the input set $P$ or the $k$-GMM $\theta$ is scaled such that $Z(\theta) \leqslant 1$, then the right hand side of equation 55 is non-negative and a $(1 + \varepsilon)$-approximation to $\phi(P, \theta)$ is indeed a $(1 + \varepsilon)$-approximation to $L(P, \theta)$. This occurs e.g. if all the eigenvalues $\sigma_{i,1}, \cdots, \sigma_{i,d}$ of $\Sigma_i$ are greater than $e^{1/d}/(2\pi)$ for every $i \in [k]$. That is, if $Z(\theta) \leqslant 1$, then for every weighted set $C'$,

$$\phi(C', \theta) \leqslant \phi(P', \theta) \leqslant (1 + \varepsilon)\phi(C', \theta)$$

implies

$$L(C', \theta) \leqslant L(P', \theta) \leqslant (1 + \varepsilon)L(C', \theta).$$

This gives the following reduction that motivates the definition of $\Theta_{d,k}$ in Section 1 that consists of $k$-GMM whose corresponding eigenvalues are at least 1. Otherwise, we multiply them by the same scaling factor (which is independent of the input $P$), and update $\lambda(\theta)$ accordingly; see Lemma C.1 and Feldman et al. (2011); Lucic et al. (2017). Such a lower bound on the eigenvalues is essential to avoid division by zero

construction of coresets of $(P, \Theta_{d,k}, \phi, \|\cdot\|_1)$ for the rest of the paper.

**Lemma C.1.** *Let $P \subseteq \mathbb{R}^d$, and $\varepsilon > 0$. Then, an $\varepsilon$-coreset of $(P, \Theta_{d,k}, \phi, \|\cdot\|_\infty)$ is an $\varepsilon$-coreset of $(P, \Theta_{d,k}, L, \|\cdot\|_\infty)$.*

*Proof.* Let $C$ be an $\varepsilon$-coreset of $(P, \Theta_{d,k}, \phi, \|\cdot\|_\infty)$. Let $\theta \in \Theta_{d,k}$, $p^* \in \arg\max_{p \in P} \phi(p, \theta)$, and $c^* \in \arg\max_{c \in C} \phi(c, \theta)$. By the definition of $C$,

$$\max_{p \in P} \phi(p, \theta) \leqslant (1 + \varepsilon)\max_{c \in C} \phi(c, \theta).$$

In particular,

$$\phi(p^*, \theta) - \phi(c^*, \theta) \leqslant \varepsilon\phi(p^*, \theta). \tag{57}$$

Therefore,

$$\max_{p \in P} L(p, \theta) - \max_{c \in C} L(c, \theta) = \max_{p \in P} \phi(p, \theta) - \ln Z(\theta) - \max_{c \in C} \left( \phi(c^*, \theta) - \ln Z(\theta) \right) \tag{58}$$

$$= \max_{p \in P} \phi(p, \theta) - \max_{c \in C} \phi(c, \theta) = \phi(p^*, \theta) - \phi(c^*, \theta) \leqslant \varepsilon \phi(p^*, \theta) \tag{59}$$

$$= \varepsilon \ln Z(\theta) + \varepsilon L(p^*, \theta) \leqslant \varepsilon \ln Z(\theta) + \varepsilon \max_{p \in P} L(p, \theta), \tag{60}$$

where the equalities in equation 58 and equation 60 hold by equation 53, and the inequality in equation 59 holds by equation 57.

For every $i \in [k]$, let $\sigma_{i,j} \geqslant 1$ denote the $j$th largest singular value of $\Sigma_i$. Ties broken arbitrarily. We then have

$$\det(2\pi\Sigma_i) = \exp(\ln \det(2\pi\Sigma_i)) = \exp\left( \ln \prod_{j=1}^{d} (2\pi\sigma_{i,j}) \right)$$

$$= \exp\left( \sum_{j=1}^{d} \ln(2\pi\sigma_{i,j}) \right)$$

$$= \exp\left( d\ln(2\pi) + \sum_{j=1}^{d} \ln(\sigma_{i,j}) \right) \geqslant \exp(d\ln(2\pi)) \geqslant e^2, \tag{61}$$

where the first inequality in equation 61 is by the assumption $\sigma_{i,j} \geqslant 1$, and the last inequality uses the assumption $d \geqslant 2$. Hence,

$$Z(\theta) := \sum_{i=1}^{k} \frac{\omega_i e}{\sqrt{\det(2\pi\Sigma_i)}} \leqslant \sum_{i=1}^{k} \frac{\omega_i e}{\sqrt{e^2}} = 1. \tag{62}$$

That is, $\ln Z(\theta) \leqslant \ln 1 = 0$, as desired. Substituting the last inequality in equation 60 yields

$$\max_{p \in P} L(p, \theta) - \max_{c \in C} L(c, \theta) \leqslant \varepsilon \ln Z(\theta) + \varepsilon \max_{p \in P} L(p, \theta) \leqslant \varepsilon \max_{p \in P} L(p, \theta).$$

Since we assume arbitrary $\theta \in \Theta_{d,k}$, $C$ is the desired coreset. $\square$

## D    FROM $k$-GMMs TO $k$-SMMs TO PROJECTIVE CLUSTERING

In this section we prove Lemma 6.2 from the main text that matches a corresponding $k$-GMM to $k$-SMM, as in Definition 6.1. Then, in Lemma D.2, we reduce the problem of fitting $k$-SMM to the input dataset $P$ to the classic problem of projective clustering in computational geometry, which is covering the points of $P$ by $k$ "slabs" (between parallel hyperplanes) of the smallest width. Recall that $\phi$ and cost were defined in equation 54 and equation 6, respectively.

**Lemma D.1** (From $k$-GMM to $k$-SMM (restatement of Lemma 6.2)). *Let $\theta \in \Theta_{d,k}$ be a $k$-GMM in $\mathbb{R}^d$. Then there is a $k$-SMM $y := (W, \omega'_1, \cdots, \omega'_k, S_1, \cdots, S_k) \in Y_{2d+1,k}$ that satisfies, for every $p \in \mathbb{R}^d$ and $p' = (p; 0; \cdots; 0) \in \mathbb{R}^{2d+1}$,*

$$W \operatorname{dist}^2(p', S(y)) \geqslant 1, \tag{63}$$

*and*

$$\phi(p, \theta) = \operatorname{cost}(p', y). \tag{64}$$

*Proof.* Identify $\theta = \left\{ (\omega_i, \Sigma_i, \mu_i) \right\}_{i=1}^{k}$. For every $i \in [k]$, let $\sigma_{i,1} > 0$ denote the largest eigenvalue of $\Sigma_i^{-1}$, and

$$W := \frac{\max_{i \in [k]} \sigma_{i,1}}{2} > 0. \tag{64}$$

Put $i \in [k]$. Since $\Sigma_i$ is a positive definite matrix, $\Sigma_i^{-1}/(2W)$ has a Cholesky Decomposition

$$A^T A = \frac{\Sigma_i^{-1}}{2W}, \tag{65}$$

where $A \in \mathbb{R}^{d \times d}$. The largest singular value of $A$ is bounded by 1, since for every unit vector $z \in \mathbb{R}^d$,

$$\|Az\|^2 = z^T A^T A z = \frac{z^T \Sigma_i^{-1} z}{2W} \leqslant \frac{\sigma_{i,1}}{2W} \leqslant 1, \tag{66}$$

where the last inequality is by equation 64.

Let $QDV^T = A$ denote the thin SVD of $A$, and $D_{j,j}$ denote the $j$th diagonal entry of $D$ (the $j$th largest singular value of $A$) for every $j \in [d]$. By equation 66 we have $D_{j,j} \leqslant 1$, so we can define the diagonal matrix $\sqrt{I - D^2} \in \mathbb{R}^{d \times d}$. Here, $\sqrt{X}$ is the matrix whose entries are squared roots of a semi positive-definite matrix $X$. By letting $L := Q\sqrt{I - D^2} \in \mathbb{R}^{d \times d}$, we have

$$I - AA^T = Q(I - D^2)Q^T = LL^T. \tag{67}$$

(Alternatively, we could compute a Cholesky decomposition of $I - AA^T$, or the SVD of $AA^T$). Similarly, let $E := I_{*,1:d} \in \mathbb{R}^{2d \times d}$ and $Y := I_{*,d+1:2d} \in \mathbb{R}^{2d \times d}$ denote, respectively the first and last $d$ columns of the identity $2d \times 2d$ matrix. Therefore,

$$I - EE^T = YY^T. \tag{68}$$

Let $B := [E \mid Y] \in \mathbb{R}^{2d \times 2d}$. Since $BB^T = EE^T + YY^T = I$, where the last equality holds by equation 68, we have that $B$ is a squared matrix whose rows are orthogonal, i.e., an orthogonal matrix where $B^T B = I$. By defining $U := B[A \mid L]^T \in \mathbb{R}^{2d \times d}$, we obtain

$$U^T U = [A \mid L] B^T B [A \mid L]^T = [A \mid L][A \mid L]^T = AA^T + LL^T = I,$$

where the last equality holds by equation 67. Moreover,

$$U^T E = U^T B I_{*,1:d} = [A \mid L] B^T B I_{*,1:d} = [A \mid L] I_{*,1:d} = A. \tag{69}$$

Let $i \in [k]$, and recall that $\mu_i$ is the mean of the $i$th $k$-GMM, and let $T_i \subseteq \mathbb{R}^{2d}$ denote the $d$-dimensional subspace that is spanned by the columns that are orthogonal to $U$, and $T_i + E\mu_i := \{t + E\mu_i \mid t \in T_i\}$ denote its translation by $E\mu_i$. By letting $p \in \mathbb{R}^d$, we have

$$\frac{1}{2}(p - \mu_i)^T \Sigma_i^{-1}(p - \mu_i) = W(p - \mu_i)^T A^T A(p - \mu_i) \tag{70}$$

$$= W\|A(p - \mu_i)\|^2 = W\|U^T E(p - \mu_i)\|^2 \tag{71}$$

$$= W\mathrm{dist}^2(E(p - \mu_i), T_i) = W\mathrm{dist}^2(Ep, T_i + E\mu_i), \tag{72}$$

where equation 70 is by equation 65, equation 71 is by equation 69, and equation 72 holds by the definition of $T_i$.

Finally, let $\xi := 1$, and add another entry to every vector $t$ in the affine subspace $T_i + E\mu_i$,

$$S_i := \left\{ \left( t; \sqrt{\frac{\xi}{W}} \right) \mid t \in T_i + E\mu_i \right\} \subseteq \mathbb{R}^{2d+1}.$$

By letting $p' := (Ep; 0) \in \mathbb{R}^{2d+1}$, we obtain

$$\mathrm{dist}^2(Ep, T_i + E\mu_i) + \frac{\xi}{W} = \mathrm{dist}^2(Ep, T_i + E\mu_i) + \left\| (0; \cdots; 0) - \left( 0; \cdots; 0; \sqrt{\frac{\xi}{W}} \right) \right\|^2 \tag{73}$$

$$= \mathrm{dist}^2(\begin{pmatrix} Ep \\ 0 \end{pmatrix}, S_i) = \mathrm{dist}^2(p', S_i),$$

where the first equality holds by the Pythagorean Theorem. Hence,

$$\frac{1}{2}(p - \mu_i)^T \Sigma_i^{-1}(p - \mu_i) + \xi = W\mathrm{dist}^2(Ep, T_i + E\mu_i) + \xi = W\mathrm{dist}^2(p', S_i), \tag{74}$$

where the first equality is by equation 72, and the second one holds after multiplying equation 73 by $W$. This proves equation 63, as

$$W\mathrm{dist}^2(p', S(y)) = \min_{i \in [k]} W\mathrm{dist}^2(p', S_i)$$

$$= \min_{i \in [k]} \frac{1}{2}(p - \mu_i)^T \Sigma_i^{-1}(p - \mu_i) + \xi \tag{75}$$

$$\geqslant \xi,$$

where equation 75 holds by equation 74, and the last inequality holds since $\Sigma$ is positive semi-definite.

Recall that $\omega_i' \geqslant 0$ was defined for every $\omega_i$ and $\Sigma_i$ in equation 52. The $k$-SMM $y = (W, \omega_1', \cdots, \omega_k', S_1, \cdots, S_k)$ satisfies the lemma as

$$\phi(p, \theta) = -\ln \sum_{i=1}^{k} \omega_i' \exp\left(-\frac{1}{2}(p - \mu_i)^T \Sigma_i^{-1}(p - \mu_i) - \xi\right)$$

$$= -\ln \sum_{i=1}^{k} \omega_i' \exp(-W \mathrm{dist}^2(p', S_i)) = \mathrm{cost}(p', y),$$

where the first equality is by equation 54, and the second is by equation 74. $\qquad\square$

**From $k$-SMMs to Projective Clustering.**  In Lemma C.1, we reduce the problem of approximating $L$ to approximating $\phi$, and in Lemma D.1 we reduce the approximation of $\phi$ to $\mathrm{cost}$. We now reduce the problem of approximating $\mathrm{cost}$ to the problem of approximating the query space $(P', Y, f, \mathrm{loss})$ of projective clustering as explained in Section 1.

Recall that $\mathrm{cost}_\infty$ and $\mathrm{dist}_\infty$ were defined in equation 9 and equation 10, respectively. The following lemma proves that $\mathrm{cost}_\infty$ for $k$-GMM is an upper bound for $W\mathrm{dist}_\infty$ of the corresponding $k$-subspaces. It will be used in our main result later.

**Lemma D.2** ($\mathrm{cost}_\infty$ upper bounds $\mathrm{dist}_\infty$).  *For every $k$-SMM $y = (W, \omega_1, \cdots, \omega_k, S_1, \cdots, S_k)$ and a finite set $P$ of points in $\mathbb{R}^d$, we have*

$$\mathrm{cost}_\infty(P, y) \geqslant W\mathrm{dist}_\infty^2(P, S(y)).$$

*Proof.* We have

$$\mathrm{cost}_\infty(P, y) = \max_{p \in P} \mathrm{cost}(p, y) \tag{76}$$

$$= \max_{p \in P} -\ln \sum_{i=1}^{k} \omega_i \exp(-W\mathrm{dist}^2(p, S_i)) \tag{77}$$

$$\geqslant \max_{p \in P} -\ln \sum_{i=1}^{k} \omega_i \exp(-W\mathrm{dist}^2(p, S(y))) = \max_{p \in P} -\ln \exp\left(-W\mathrm{dist}^2(p, S(y))\right) \tag{78}$$

$$= W\mathrm{dist}_\infty^2(P, S(y)), \tag{79}$$

where equation 76 and equation 77 hold by definition, and equation 78 holds since $\mathrm{dist}(p, S_i) \geqslant \mathrm{dist}(p, S(y)) = \max_{i \in [k]} \mathrm{dist}(p, S_i)$. $\qquad\square$

# E  FROM SENSITIVITY TO $\ell_\infty$-CORESET

In order to bound sensitivities, we generalize the reduction that was suggested in Varadarajan and Xiao (2012a) to compute sensitivities via $\ell_\infty$ coreset, where instead of approximating the sum of fitting costs or distances, we approximate their maximum. We will then use it to bound sensitivities of $k$-GMMs via projective clustering. In this section we assume that $(P, Y, \mathrm{cost}, \mathrm{loss})$ is a general query space, i.e., $\mathrm{cost}$ may be any real function on $P \times Y$, and $Y$ and $P$ are arbitrary sets (not necessarily points or even in an Euclidean space).

## E.1  NON-WEIGHTED INPUT

The original reduction from sensitivity to $\ell_\infty$ coreset in Agarwal et al. (2005) was for a specific problem and for non-weighted data. For simplicity and intuition, we first generalize it to any query space and only then reduce non-unit weights to positive weights, following the ideas in Feldman et al. (2013). The cardinality $n$ of the input set $P$ is replaced by the sum of input weights. Since we are

interested in multiplicative error factors, it would be useful to normalize this sum and define, for a weighted set $C' := (C, w)$,

$$\overline{w}(C') := \frac{\sum_{c \in C} w(c)}{\min_q w(q)},$$

where the minimum is over every $q \in C$ such that $w(c) > 0$. If this minimum is 0, we define $\overline{w}(C') := 0$. We now use the following definition of a coreset scheme as an algorithm that computes coresets.

**Definition E.1** (coreset scheme). *Let $(P, Y, \mathrm{cost}, \mathrm{loss})$ be a query space such that $P$ is an unweighted set. Let* $\mathrm{size} : [0, \infty)^4 \to [1, \infty)$ *and* $\mathrm{time} : [0, \infty)^4 \to [0, \infty)$ *be a pair of functions. Let* $\mathrm{CORESET}(Q', \varepsilon, \delta)$ *be an algorithm that gets as input a weighted set $Q' = (Q, w)$ such that $Q \subseteq P$, an approximation error $\varepsilon > 0$ and a probability $\delta \in (0, 1)$ of failure. The tuple $(\mathrm{CORESET}, \mathrm{size}, \mathrm{time})$ is called an $(\varepsilon, \delta)$-coreset scheme for $(P, Y, \mathrm{cost}, \mathrm{loss})$ and some $\varepsilon, \delta > 0$, if (i)-(iii) hold as follows:*

*(i) A call to $\mathrm{CORESET}(Q', \varepsilon, \delta)$ returns a weighted set $(C, u)$.*

*(ii) With probability at least $1 - \delta$, $(C, u)$ is an $\varepsilon$-coreset of $(Q', Y, \mathrm{cost}, \mathrm{loss})$.*

*(iii) $(C, u)$ can be computed in $\mathrm{time}(|Q|, \overline{w}(Q'), \varepsilon, \delta)$ time and its size is*

$$|C| \leqslant \mathrm{size}(|Q|, \overline{w}(Q'), \varepsilon, \delta).$$

---

**Algorithm E.2:** $\mathrm{SENSITIVITY}(P, \varepsilon, \delta, \ell_\infty\text{-}\mathrm{CORESET})$; see Theorem E.7

**Input:** A finite set $P \subseteq \mathbb{R}^d$, an approximation error $\varepsilon > 0$, and probability $\delta \in (0, 1)$ of failure.
**Required:** A coreset scheme $\ell_\infty\text{-}\mathrm{CORESET}$ for $(P, Y, \mathrm{cost}, \|\cdot\|_\infty)$.
**Output:** A sensitivity bound $s : P \to (0, \infty)$.

1   $P_1 := P; \quad i := 1$
2   **while** $|P_i| \geqslant 1$ // $P_i$ is not an empty set
3   **do**
4      $S_i := \ell_\infty\text{-}\mathrm{CORESET}(P_i, \varepsilon, \delta/|P|)$
5      **for** *every $p \in S_i$* **do**
6         $s(p) := \dfrac{(1 + \varepsilon)}{i}$
7      $P_{i+1} := P_i \backslash S_i$
8      $i := i + 1$
9   **return** $s$

---

The following result shows how a coreset scheme for $\ell_\infty$ coresets can be used to bound sensitivity. The total sensitivity depends on the size of the $\ell_\infty$ coreset, which in turn determines the size of the desired $\ell_1$ coreset. It is a variant, improvement and generalization of (Feldman et al., 2013, Lemma 49) which is in turn a variant of (Varadarajan and Xiao, 2012a, Lemma 3.1).

Recall that sensitivity bound and total sensitivity were defined in Definition B.4.

**Lemma E.3** (generalization of Feldman et al. (2013); Varadarajan and Xiao (2012a)). *Let $P$ be a set of size $n = |P|$, and $(\ell_\infty\text{-}\mathrm{CORESET}, \mathrm{size}, \mathrm{time})$ be a coreset scheme for $(P, Y, \mathrm{cost}, \|\cdot\|_\infty)$. Let $\varepsilon, \delta \in (0, 1/2)$, and let $s : P \to (0, \infty)$ be the output of a call to $\mathrm{SENSITIVITY}(P, \varepsilon, \delta, \ell_\infty\text{-}\mathrm{CORESET})$; See Algorithm E.2. Then, with probability at least $1 - \delta$, $s$ is a sensitivity bound for $(P, Y, \mathrm{cost})$ whose total sensitivity is*

$$\sum_{p \in P} s(p) \in \mathrm{size}(n, n, \varepsilon, \delta/n)(1 + \varepsilon) O(\log n). \tag{80}$$

*Moreover, the function $s$ can be computed in $O(n) \cdot \mathrm{time}(n, n, \varepsilon, \delta/n)$ time.*

*Proof.* **Probability of failure.** For $i \in [n]$, the event that $S_i$ is an $\varepsilon$-coreset for $(P_i, Y, \mathrm{cost}, \|\cdot\|_\infty)$ during the execution of Line 4 occurs with probability at least $1 - \delta/n$. For the rest of the proof, suppose that this event indeed occurred for every $i \in [n]$, which happens with probability at least $1 - \delta$, by the union bound.

**Correctness.** We first bound the total sensitivity and then the computation time of $s$. Algorithm E.2 implements the algorithm that is described in the following proof. Let $i := 1$, and $P_1 := P$. By its construction, $|S_i| \leqslant \text{size}(n, n, \varepsilon, \delta)$, and for every $y \in Y$

$$\max_{p \in P_i} \text{cost}(p, y) \leqslant (1 + \varepsilon) \max_{p \in S_i} \text{cost}(p, y).$$

Recursively define $P_{i+1} := P_i \backslash S_i$ for every non-empty set $P_i$. Hence, $|P_{i+1}| < |P_i|$, and thus $P_\ell = \varnothing$ for

$$\ell \leqslant |P_1| = n. \tag{81}$$

Let $p \in P$, and $v(p) \in [\ell]$ such that $p \in S_{v(p)}$. Let $y \in Y$ such that $\text{cost}(p, y) > 0$. Finally, let $j \in [v(p)]$, and $s_j \in \arg\max_{s \in S_j} \text{cost}(s, y)$ denote the "farthest point" in $S_j$ from $y$. Since $p \in P_j$,

$$\text{cost}(p, y) \leqslant \max_{q \in P_j} \text{cost}(q, y) \leqslant (1 + \varepsilon) \max_{s \in S_j} \text{cost}(s, y) = (1 + \varepsilon)\text{cost}(s_j, y), \tag{82}$$

where the second inequality holds by the definition of $S_j$. We can now bound $\text{cost}(p, y)$ by

$$\text{cost}(p, y) \leqslant (1 + \varepsilon) \min_{j \in [v(p)]} \text{cost}(s_j, y) \tag{83}$$

$$\leqslant (1 + \varepsilon) \cdot \frac{\sum_{j=1}^{v(p)} \text{cost}(s_j, y)}{v(p)} \tag{84}$$

$$\leqslant \frac{(1 + \varepsilon)}{v(p)} \sum_{p \in P} \text{cost}(p, y), \tag{85}$$

where equation 83 holds by equation 82, equation 84 holds since the minimum cannot be larger than the average, and equation 85 holds since $\{s_1, \cdots, s_v\} \subseteq P$. Hence,

$$s(p) := \frac{(1 + \varepsilon)}{v(p)} \geqslant \frac{\text{cost}(p, y)}{\sum_{q \in P} \text{cost}(q, y)}, \tag{86}$$

where the inequality holds by equation 83. Since $y$ is an arbitrary query, $s(p)$ is a sensitivity bound, as desired by the lemma and Definition B.4. Summing equation 86 over $p \in P$ bounds the total sensitivity

$$\sum_{p \in P} s(p) = (1 + \varepsilon) \sum_{p \in P} \frac{1}{v(p)}$$

$$= (1 + \varepsilon) \sum_{j=1}^{\ell} \frac{|S_j|}{j} \leqslant (1 + \varepsilon) \sum_{j=1}^{n} \frac{\text{size}(n, n, \varepsilon, \delta/n)}{j} \in (1 + \varepsilon)\text{size}(n, n, \varepsilon, \delta/n)O(\log n),$$

where the equality holds since $|S_j|$ points were removed from $P_j$ during the $j$th iteration of the algorithm, and each $p \in S_j$ was labeled $v(p) = j$, for every $j \in [\ell]$. The last deviation holds by the definition of size equation 81 and the fact that $\sum_{i=1}^{n} 1/n = O(\log n)$ is an harmonic sequence for every integer $n \geqslant 1$; see Theorem E.5 for exact bound.

**Running time.** For every $i \in [\ell]$, computing $S_i$ (whose cardinality is at most $n$) for $P_i$ takes at most $\text{time}(n, n, \varepsilon, \delta/n)$ time. By equation 81, computing all the $n$ sets takes $n \cdot \text{time}(n, n, \varepsilon, \delta/n)$ time. Removing $S_i$ from $B_i$ (e.g. using linked lists), as well as computing the values of $s$, takes $O(n)$ time, so the dominated time is $n \cdot \text{time}(n, n, \varepsilon, \delta/n)$. $\qquad \square$

### E.2 WEIGHTED INPUT

In this section we generalize the result of the previous section to non-weighted input. This is a generalization of the idea that was suggested in Feldman et al. (2013) with little better bounds.

The following constant for approximating harmonic sequences can be approximated very efficiently, in exponential convergence rate. However, in the next corollary we use it only for the analysis, since we use it to compute the difference between two harmonic sequences.

---

**Algorithm E.4:** WSENSITIVITY$_{\ell_\infty}$-CORESET$(P', \varepsilon, \delta, \ell_\infty$-CORESET$)$; see Theorem E.7.

---

**Input:** A weighted set $P' = (P, w)$ of points in $\mathbb{R}^d$, an approximation error $\varepsilon > 0$,
probability of failure $\delta \in (0, 1)$, and
a coreset scheme $\ell_\infty$-CORESET for $(P, Y, \text{cost}, \|\cdot\|_\infty)$; see Definition E.1.

**Output:** $s : P \to (0, \infty)$

1   $w_{\min} = \min_{p \in P} w(p)$

2   **for** *every* $p \in P$ **do**

3      $h(p) := \left\lceil \frac{w(p)}{\varepsilon w_{\min}} \right\rceil$

4      $s(p) := 0$

5   $P_1 := P; \quad i := 1$

6   **while** $|P_i| \geqslant 1$ **do**

7      $S_i := \ell_\infty$-CORESET$(P_i, \varepsilon, \frac{\delta}{|P_1|})$

8      Set $q_i \in \arg\min_{p \in S_i} h(p)$

9      **for** *every* $p \in S_i$ **do**

10        $h(p) := h(p) - h(q_i)$

11        $m := i + h(q_i) - 1$

12        $s(p) := s(p) + (1 + \varepsilon)^2 \left( \frac{1}{i_j} + \ln\left( \frac{m_j}{i_j} \right) + \frac{1}{2m_j} - \frac{1}{2i_j} + \frac{1}{i_j^2} \right)$

         `//  ~ s(p) + (1 + ε)² Σ`$_{j=i}^m$ `1/j;` `see Corollary E.6`

13      $P_{m+1} := P_i \backslash \{q_i\}$

14      $i := m + 1$

15   **return** $s$

---

**Theorem E.5** (Euler–Mascheroni Constant Mortici (2010)). *Let $n \geqslant 1$ be an integer. Then there is a constant $\gamma$ (independent of $n$) such that*

$$0 < \ln n + \gamma + \frac{1}{2n} - \sum_{i=1}^n \frac{1}{i} \leqslant \frac{1}{n^2}.$$

**Corollary E.6.** *For every pair $(m, i)$ of integers that satisfy $m \geqslant i \geqslant 2$,*

$$-\frac{1}{(i-1)^2} < \ln\left( \frac{m}{i-1} \right) + \frac{1}{2m} - \frac{1}{2(i-1)} - \sum_{j=i}^m \frac{1}{j} \leqslant \frac{1}{m^2} \tag{87}$$

*Proof.* For the left hand side of equation 87, we have by Theorem E.5

$$\sum_{j=i}^m \frac{1}{j} = \sum_{j=1}^m \frac{1}{j} - \sum_{j=1}^{i-1} \frac{1}{j} < \ln m + \gamma + \frac{1}{2m} - \left( \ln(i-1) + \gamma + \frac{1}{2(i-1)} - \frac{1}{(i-1)^2} \right)$$

$$= \ln\left( \frac{m}{i-1} \right) + \frac{1}{2m} - \frac{1}{2(i-1)} + \frac{1}{(i-1)^2}.$$

Similarly,

$$\sum_{j=i}^m \frac{1}{j} = \sum_{j=1}^m \frac{1}{j} - \sum_{j=1}^{i-1} \frac{1}{j} \geqslant \ln m + \gamma + \frac{1}{2m} - \frac{1}{m^2} - \left( \ln(i-1) + \gamma + \frac{1}{2(i-1)} \right)$$

$$= \ln\left( \frac{m}{i-1} \right) + \frac{1}{2m} - \frac{1}{2(i-1)} - \frac{1}{m^2}.$$

$\square$

**Theorem E.7** (Varadarajan and Xiao (2012a); Feldman et al. (2013)). *Let $(P, w)$ be a positively weighed set of size $n = |P|$, and $(\ell_\infty$-CORESET, size, time) be a coreset scheme for $(P, Y, \text{cost}, \|\cdot\|_\infty)$. Let $\varepsilon, \delta \in (0, 1]$, and $s : P \to (0, \infty)$ be the output of a call to*

WSENSITIVITY$_{\ell_\infty}$-CORESET$((P, w), \varepsilon, \delta, \ell_\infty$-CORESET$)$;*See Algorithm E.4. Then, with probability at least $1 - \delta$, $s$ is a sensitivity bound for $((P, w), Y, \mathrm{cost})$, its total sensitivity is*

$$\sum_{p \in P} s(p) \in \mathrm{size}(n, n, \varepsilon, \delta/n) O\left(\log \frac{\overline{w}(P)}{\varepsilon}\right),$$

*and the function $s$ can be computed in $O(n) \cdot \mathrm{time}(n, n, \varepsilon, \delta/n)$ time.*

*Proof.* **The probability** that the construction of the coreset in Line 7 would succeed during all the $n$ iterations of the algorithm is at least $1 - \delta$, by the union bound.

**Sensitivity bound:** Let $p \in P$ and consider the values of $h$, $s$ and $w_{\min}$ during the end of the execution of Algorithm E.4. We have

$$x \leqslant y \cdot \left\lceil \frac{x}{y} \right\rceil \leqslant y \left( \frac{x}{y} + 1 \right) = x + y$$

for every $x, y > 0$. Substituting $x = w(p)$, $y = \varepsilon w_{\min}$ and $h(p) = \lceil x/y \rceil$, yields

$$w(p) \leqslant \varepsilon w_{\min} h(p) = y\lceil x/y \rceil \leqslant x + y = w(p) + \varepsilon w_{\min} \leqslant (1 + \varepsilon)w(p). \tag{88}$$

Hence, by letting $s_h^*(p)$ denote the sensitivity of $((P, h), Y, \mathrm{cost})$, we obtain

$$\begin{aligned}
s^*(p) &:= \sup_{y \in Y : \mathrm{cost}(p,y) > 0} \frac{w(p)\mathrm{cost}(p, y)}{\sum_{q \in P} w(q)\mathrm{cost}(q, y)} \\
&\leqslant \sup_{y \in Y : \mathrm{cost}(p,y) > 0} \frac{\varepsilon w_{\min} h(p)\mathrm{cost}(p, y)}{\sum_{q \in P} \varepsilon w_{\min} h(q)\mathrm{cost}(q, y)/(1 + \varepsilon)} = (1 + \varepsilon)s_h^*(p),
\end{aligned} \tag{89}$$

where the inequality holds by equation 88. It is left to bound $s_h^*(p)$. Let $Q$ denote the (unweighted) multi-set that consists of $h(p)$ copies of every point $p \in P$. Let $s' : Q \to [0, \infty)$ denote the output of a call to SENSITIVITY$(Q, \varepsilon, \delta, \ell_\infty$-CORESET$)$; see Algorithm E.2. By Lemma E.3, for a single copy of a point $p$ in $Q$ we have, with probability at least $1 - \delta$,

$$s'(p) \geqslant \sup_{y \in Y, \mathrm{cost}(p,y) > 0} \frac{\mathrm{cost}(p, y)}{\sum_{q \in P} h(q)\mathrm{cost}(q, y)}$$

so for all its $h(p)$ copies we have

$$h(p)s'(p) \geqslant \sup_{y \in Y, \mathrm{cost}(p,y) > 0} \frac{h(p)\mathrm{cost}(p, y)}{\sum_{q \in P} h(q)\mathrm{cost}(q, y)} = s_h^*(p). \tag{90}$$

That is, $s_h^*(p) \leqslant h(p)s'(p)$. Next, we bound $h(p)s'(p)$ by $s(p)$.

Note that the number of copies of a point $p$ in $Q$ has no effect on the computation of the coreset $S_1$ during the first iteration of the call to SENSITIVITY$(Q, \varepsilon, \delta, \ell_\infty$-CORESET$)$, so $S_1$ may be computed on the *unweighted set* of the $n$ distinct points in $Q$. Moreover, this number $n$ of distinct points will remain the same after the first iteration, as well as the following coresets $S_2, S_3, \cdots$, until all the copies of some point $q_1 \in S_1$ will be removed in Line 7 of SENSITIVITY. This point $q_1$ is the point with the smallest number of duplicates (weights) in $S_1$. During these $h(q_1)$ iterations, the value $s'(p)$ of (one of the copies of) each $p \in S_1$ is increased in Line 6 by,

$$\frac{1 + \varepsilon}{1} + \cdots + \frac{1 + \varepsilon}{h(q_1)} = (1 + \varepsilon) \sum_{j=1}^{h(q_1)} \frac{1}{j} < (1 + \varepsilon)\left( \ln\left( \frac{h(q_1)}{1} \right) + \frac{1}{2h(q_1)} + \frac{1}{2} + 1 \right),$$

where the last inequality is by substituting $i := 2$ and $m := h(q_1)$ in the left hand side of Corollary E.6. This is indeed the value that is added to $s(p)$ in Line 12 of Algorithm E.4.

Similarly, during the $i$th iteration $q_i \in S_i$ is the point with the smallest number of remaining copies in $Q = P_i$. In Algorithm E.4, $q_i$ is removed in its $j$th iteration for some $j \in [n]$. Let $i_j$ and $m_j$ respectively denote the value of $i$ and $m$ during the execution of the $j$th iteration. Hence, $i_1 = 1$, $i_{j+1} = i_j + h(q_{i_j})$ and $m_j = i_j + h(q_{i_j}) - 1 = i_{j+1} - 1$, for every $j \in [n]$. We obtain that $S_i$ is the

same for every iteration $i \in [i_j, m_j]$ in Algorithm E.2. During these $h(q_{i_{j-1}+1})$ iterations, until $q_{i_j}$ is removed from $Q = P_{i_j}$, the value $s'(p)$ of every $p \in S_i$ was increased by

$$h(p)s'(p) = \frac{1+\varepsilon}{i_j} + \cdots + \frac{1+\varepsilon}{m_j} = (1+\varepsilon)\sum_{k=i_j}^{m_j}\frac{1}{k} < (1+\varepsilon)\left(\ln\left(\frac{m_j}{i_j}\right) + \frac{1}{2m_j} + \frac{1}{2i_j} + \frac{1}{i_j^2}\right),$$

where the inequality is by substituting $i := i_j + 1$ and $m := m_j$ in the left hand side of equation 87.

The right hand side of the last inequality multiplied by $(1+\varepsilon)$ is the update of $s(p)$ in Line 12 of the $j$th iteration in Algorithm E.4 that imitates the updates of $s'(p)$ during iterations $i_j$ till $i_{j+1}$ of Algorithm E.2. Hence,

$$(1+\varepsilon)h(p)s'(p) \leqslant s(p). \tag{91}$$

This proves the desired sensitivity bound as

$$s(p) \geqslant (1+\varepsilon)h(p)s'(p) \geqslant (1+\varepsilon)s_h^*(p) \geqslant s^*(p) = \sup_{y \in Y:\mathrm{cost}(p,y)>0}\frac{w(p)\mathrm{cost}(p,y)}{\sum_{q \in P}w(q)\mathrm{cost}(q,y)},$$

where the first inequality is by equation 91, the second is by equation 90, and the third is by equation 89.

**The total sensitivity** of $s$ is bounded using the fact that $\varepsilon \leqslant 1$, $m_j = i_{j+1} - 1$ and

$$\sum_{j=1}^{n}\frac{1}{i_j} + \sum_{j=1}^{n}\left(\ln\left(\frac{i_{j+1}-1}{i_j}\right) + \frac{1}{2(i_{j+1}-1)} - \frac{1}{2i_j} + \frac{1}{i_j^2}\right) \leqslant \sum_{j=1}^{n}\frac{1}{i_j} + \sum_{j=1}^{n}\frac{1}{i_{j+1}^2} + \sum_{j=1}^{n}\sum_{k=i_j+1}^{i_{j+1}-1}\frac{1}{k} \tag{92}$$

$$\leqslant \sum_{j=1}^{n}\frac{1}{i_j} + \sum_{j=1}^{\infty}\frac{1}{j^2} + \sum_{k=1}^{i_{n+1}-1}\frac{1}{k} \in O(\log(i_{n+1})) = O\left(\sum_{p \in P}h(p)\right) = O\left(\log\frac{\overline{w}(P)}{\varepsilon}\right). \tag{93}$$

where equation 92 is by substituting $i = i_j + 1$ and $m = i_{j+1} - 1$ in the right hand side of equation 87, and equation 93 holds since $2 = \sum_{j=1}^{\infty}1/j^2$, and $\sum_{k=1}^{m} \in O(\lg m)$ by equation 87. Summing the accumulated sensitivities over every $j \in [n]$ iteration, each over $|S_{i_j}|$ points yields

$$\sum_{p \in P}s(p) \in (1+\varepsilon)^2\mathrm{size}(n,n,\varepsilon,\delta/n)O\left(\log\frac{\overline{w}(P)}{\varepsilon}\right),$$

where the last derivation is by equation 93 and the definition of $S_{i_j}$ in Line 7 of Algorithm E.4.

**The running time** follows from the fact that in the $j$th "for" iteration, the point $q_{i_j}$ is removed from $P$, so there are $n$ iterations. The dominated time in each of the $n$ iterations is computing the coreset $S_i$ in $\mathrm{time}(n,n,\varepsilon,\delta/n)$ time. $\qquad\square$

By combining Theorem E.7 and Theorem B.5 we obtain the following corollary which shows how to compute coresets with $\|\cdot\|_1$ loss based on coresets for $\|\cdot\|_\infty$ loss on weighted data.

**Corollary E.8.** *Let*

- $(P, w, Y, \mathrm{cost}, \|\cdot\|_1)$ *be a query space.*

- $f : P \times Y \to [0, \infty)$ *such that for every $p \in P$ and $y \in Y$,*

$$f(p,y) = \begin{cases} \frac{\mathrm{cost}(p,y)}{\sum_{p \in P}w(p)\mathrm{cost}(p,q)} & \mathrm{cost}(p,q) > 0 \\ 0 & \mathrm{cost}(p,y) = 0, \end{cases} \tag{94}$$

- $s$, $\mathrm{time}$ *and* $\mathrm{size}$ *be defined as in Theorem E.7.*

- $\varepsilon, \delta \in (0,1)$*, and $m$ be defined as in Theorem B.5.*

- $(C, u)$ be the output of a call to CORESET$(P, w, s, m)$; see Algorithm B.6.

*Then $(i)$–$(v)$ hold as follows.*

  *(i) With probability at least $1 - \delta$, $(C, u)$ is an $\varepsilon$-coreset for $((P, w), Y, \text{cost}, \|\cdot\|_1)$.*

  *(ii) $C \subseteq P$ and $|C| \in O(m)$.*

  *(iii) $C$ can be computed in $O(n) \cdot \text{time}(n, n, \varepsilon, \delta/n))$ time where $n = |P|$.*

  *(iv) $u(p) \in \left[w(p), \sum_{q \in P} w(q)/m\right]$ for every $p \in C$, and*

  *(v) $\sum_{q \in C} u(q) = \sum_{q \in P} w(q)$.*

## F  CORESETS FOR STREAMING DATA

In the previous sections we bound the sensitivity and dimension of the desired query space $(P', Y, f, \text{loss})$. However, as stated later in Lemma G.5, the construction time of the coreset is quadratic in $n$. This is due to the computation time of the sensitivity $s$ in Corollary E.7. To obtain a near-linear time algorithm, we use the well-known streaming approach that is described in this section. It enables us to compute the coreset only on small subsets of the input $n$ times. Hence, we use it even if all the input points are given (off-line).

The idea behind the merge-and-reduce tree that is shown in Algorithm F.4 is to merge every pair of subsets and then reduce them by half. The relevant question is what is the smallest size of input that the given coreset can reduce by half. The log-Lipschitz property below is needed for approximating the cumulative error during the construction of the tree. In the following definition "sequence" is an ordered multi-set.

**Definition F.1** (input stream). *Let $P$ be a (possibly infinite, unweighted) set. A stream of points from $P$ is a procedure whose $i$th call returns the $i$th points $p_i$ in a sequence $(p_1, p_2, \cdots)$ of points that are contained in $P$, for every $i \geqslant 1$.*

**Definition F.2** (halving function). *Let $\varepsilon, \delta, r > 0$. A non-decreasing function $s : [0, \infty) \to [0, \infty)$ is an $(\varepsilon, \delta, r)$-halving function of a function* size $: [0, \infty)^4 \to [0, \infty)$ *if for every integer $h \geqslant 1$, $n = s(h)$, and $w' = 2^h n$ we have*

$$\text{size}(2n, w', \varepsilon/h, \delta/4^h) \leqslant n,$$

*and $s$ is $r$-log-Lipschitz over $[c, \infty)$ for some $c = O(1)$, i.e., for every $\Delta, h \geqslant c$ we have $s(\Delta h) \leqslant \Delta^r s(h)$.*

**Definition F.3** (mergable coreset scheme). *Let $(\text{CORESET}, \text{size}, \text{time})$ be an $(\varepsilon, \delta)$-coreset scheme for the query space $(P, Y, \text{cost}, \text{loss})$, such that the total weight of the coreset and the input is the same, i.e. a call to CORESET$((Q, w), \varepsilon, \delta)$ returns a weighted set $(C, u)$ whose overall weight is $\sum_{p \in C} u(p) = \sum_{p \in Q} w(p)$. Let $s$ be an $(\varepsilon, \delta, r)$-halving function for* size. *Then the tuple $(\text{CORESET}, s, \text{time})$ is an $(\varepsilon, \delta, r)$-mergable coreset scheme for $(P, Y, \text{cost}, \text{loss})$.*

The following theorem states a reduction from off-line coreset construction to a coreset that is maintained during streaming. The required memory and update time depends only logarithmically in the number $n$ of points seen so far in the stream. It also depends on the halving function that corresponds to the coreset via $s(\cdot)$.

The theorem below holds for a specific $n$ with probability at least $1 - \delta$. However, by the union bound we can replace $\delta$ by, say, $\delta/n^2$ and obtain, with high probability, a coreset $C'_n$ for each of the $n$ point insertions, simultaneously.

**Theorem F.5** (generalization of Feldman et al. (2013)). *Let $(\text{CORESET}, s, \text{time})$ be an $(\varepsilon, \delta)$-mergable coreset scheme for $(P, Y, \text{cost}, \text{loss})$ and $s$ be its $(\varepsilon, \delta, r)$-halving function of size $r \geqslant 1$ is constant, and $\varepsilon, \delta \in (0, 1/2)$. Let $stream$ be a stream of points from $P$. Let $C'_n$ be the $n$th output weighted set of a call to STREAMING$_{\text{CORESET}}(stream, \varepsilon/6, \delta/6, \text{CORESET}, s)$; see Algorithm F.4. Then, with probability at least $1 - \delta$,*

  *(i) (Correctness) $C'_n$ is an $\varepsilon$-coreset of $(P_n, Y, \text{cost}, \text{loss})$, where $P_n$ is the first $n$ points in stream.*

(a) Construct a coreset $S_0$ of size $|S_0| = m/2$ from the first $m$ points in the stream.

(b) Read the next $m$ points, merge their coreset with $S_0$ to obtain $S_1$.

(c) Read the next $m$ points, construct a coreset $S_0$ of size $|S_0| = m/2$.

(d) Read the next $m$ points, merge their coreset with $S_0$ then with $S_1$ to obtain $S_2$.

(e) Read the next $m$ points, construct a coreset $S_0$ of size $|S_0| = m/2$.

(f) Read the next $m$ points, merge their coreset with $S_0$ to obtain $S_1$.

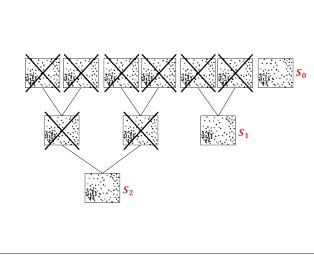

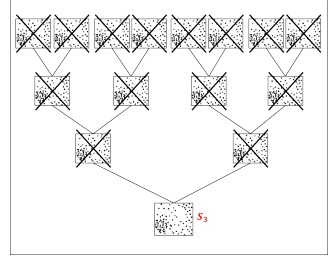

(g) Read the next $m$ points, construct a coreset $S_0$ of size $|S_0| = m/2$.

(h) Read the next $m$ points, merged their coreset with $S_0$ then with $S_1$ then with $S_2$ to obtain $S_3$.

Figure 4: Illustration of Algorithm F.4. Let $P$ be a set of $m$ points, error parameter $\varepsilon \in (0, \frac{1}{2})$ and probability of success $\delta \in (0, \frac{1}{2})$. Assume a coreset function $f(P, \varepsilon, \delta)$ returns a coreset of size $\frac{1}{2}m$ with $\varepsilon$ error parameter and $\delta$ probability of success. This figure shows an algorithm for $n = 8m$ streaming points. The algorithm maintains a binary tree, where each new $n$ points are added to the tree as a leaf. Every two nodes with the same level are merged using the coreset function $f$ to a node in next level. Hence, each level has maximum of one node, and a total of $\mathcal{O}(\log \frac{n}{m})$ nodes.

---

**Algorithm F.4:** $\text{STREAMING}_{\text{CORESET}}(stream, \varepsilon, \delta, \text{CORESET}, s)$

---

**Input:**    An input $stream$ of points from a set $P$,
                 an error parameter $\varepsilon \in (0, 1/2)$, probability of success $\delta \in (0, 1/2)$, and

**Required:** An algorithm $\text{CORESET}$ and $s : [0, \infty) \to [0, \infty)$ such that $(\text{CORESET}, s, \text{time})$ is a
                 mergable coreset scheme for $(P, Y, \text{cost}, \text{loss})$.

**Output:**  A sequence $C'_1, C'_2, \cdots$ of coresets that satisfies Theorem F.5.

1   **for** *every integer $h$ from $1$ to $\infty$* **do**
2      Set $S_i := \varnothing$ for every integer $i \geqslant 0$
3      $T_{h-1} \leftarrow S_{h-1}$
4      **for** $2^{h-1} \cdot s(h)$ *iterations* **do**
5          Read the next point $p$ in $stream$ and add it to $S_0$
6          **if** $|S_0| = s(h)$ **then**
7              $i := 0; S := \varnothing$
8              **while** $S_i \neq \varnothing$ **do**
9                  $S := \text{CORESET}\left(S \cup S_i, \frac{\varepsilon}{h}, \frac{\delta}{4^h}\right)$
10                 $S_i := \varnothing$
11                 $i := i + 1$
12              $S_i := S$
13          $C'_n := \text{CORESET}\left(\left(\bigcup_{i=0}^{h-1} T_i\right) \cup \left(\bigcup_{i=0}^{h} S_i\right), \varepsilon, \delta\right)$
14          **Output** $C'_n$

---

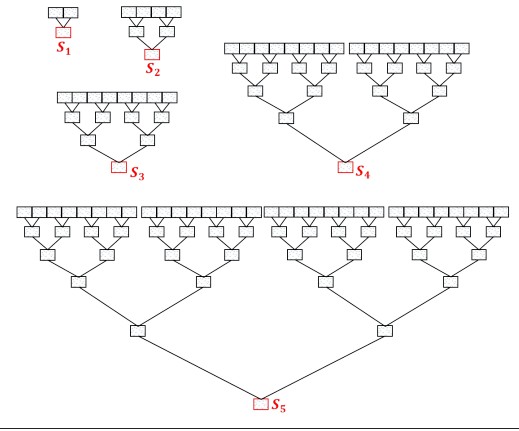

Figure 5: Using coreset function on a corest increase the error. Therefore when building the tree, instead of using input $\varepsilon$, we use a scaled down $\varepsilon$ with respect to the tree hight. However, when stream size is unknown (or $\infty$) the tree height is also unknown. So, we use several trees with height between $1$ and $\infty$. Here we can see five tress, where for each tree we store only the coreset at the head.

*(ii) (Size)* $|C_n| \in \text{size}(s(c), n, \varepsilon, \delta)$ *for some constant* $c \geqslant 1$.

*(iii) (Memory) there are at most* $b = s(c) \cdot O(\log^{r+1} n)$ *points in memory during the computation of* $C'_n$.

*(iv) (Update time)* $C'_n$ *is outputted in additional* $t \in O(\log n) \cdot \text{time}(b, n, \frac{\varepsilon}{O(\log n)}, \frac{\delta}{n^{O(1)}})$ *time after* $C'_{n-1}$.

*(v) (Overall time)* $C'_n$ *is computed in* $nt$ *time.*

*Proof.* We prove that for a call to $\text{STREAMING}_{\text{CORESET}}(stream, \varepsilon, \delta, \text{CORESET})$, the theorem holds if we replace $\varepsilon$ with $6\varepsilon$ in properties $(i)$ to $(v)$, and $\delta$ with $6\delta$. This would prove the theorem for a call to $\text{STREAMING}_{\text{CORESET}}(stream, \varepsilon/6, \delta/6, \text{CORESET})$.

Let $h \geqslant 1$, and let $P_h$ denote the set of points that are read from $stream$ during the $h$th "for" iteration in Line 1. We consider the values of $h$, $S_i$, and $T_h$ during the time that $C'_n$ was outputted, after reading the first $n$ points from the stream. We define $s(h)$ as in Definition F.2.

The set $P_h$ can be partitioned into equal consecutive $2^{h-1} = |P_h|/s(h)$ subsets $P_{h,1}, P_{h,2}, \cdots, P_{h,m}$, each of size $s(h)$ by Line 4. We now recursively define a binary complete and full tree that corresponds to $P_h$ whose height is $h$ levels, where each of its nodes corresponds to an $(\varepsilon/h)$-coreset $S$ that is computed, with probability at least $1 - \delta/4^h$, in Line 9; see Fig. 4. The $j$th leftmost leaf of the tree for $P_h$, for every $j \in [2^{h-1}]$, is the $(\varepsilon/h)$-coreset of $P_{h,j}$. An inner node in the $i$th level corresponds to the $(\varepsilon/h)$-coreset of the union $S_i$ of coresets that correspond to its pair of children and their corresponding input points. Hence, the root of this tree corresponds to the coreset $S_h = T_h$ of $P_h$; see Line 3.

The input to each coreset construction call is therefore the union $C_1 \cup C_2$ of a pair of weighted sets. In the leaves, each coreset has size of at most $s(h)/2$ points, due to the definition of the halving function, so the input to the second level is of size $|C_1 \cup C_2| \leqslant s(h)$ points. The sum of weights in $C_1 \cup C_2$ equals to the number of input points they represent, by Definition F.3 of a mergeable coreset, and it is at most $W = |P_h| = 2^{h-1}s(h)$. The output coreset has therefore size at most $s(h) - 1$ or $s(h)/2$, if $|C_1 \cup C_2| \leqslant s(h) - 1$ or $|C_1 \cup C_2| = s(h)$, respectively. Similarly, in the higher levels, the input is a union of coresets, each of size at most $s(h) - 1$, which is also an upper bound on the size of the output coreset.

**The probability** that a coreset call fails during the construction of a coreset for the tree of $T_h$ in Line 9 is $\delta/4^h$, and the number of such calls is the number $2^h - 1$ of nodes in this tree. Using the union bound, one of these constructions will fail with probability at most

$$\frac{\delta}{4^h} \cdot (2^h - 1) < \frac{\delta}{2^h} \leqslant \frac{2\delta}{h^2}.$$

The probability that one of the coreset during the construction of all the trees in the stream would fail is thus by the union bound,

$$\sum_{h=1}^{\infty} \frac{2\delta}{h^2} = 2\delta \sum_{h=1}^{\infty} \frac{1}{h^2} \leqslant 4\delta. \tag{95}$$

Suppose that all the coreset constructions in Line 9 indeed succeed (which happens with probability at least $1 - 4\delta$). In particular, the input to CORESET in Line 13 is a union of coresets. The construction of $C'_n$ in Line refforh12 would fail with probability at most $\delta$, and thus $C'_n$ is an $\varepsilon$-coreset with probability at least $1 - 5\delta \geqslant 1 - 6\delta$.

**The required memory** for storing the $O(h)$ coresets $S_0, \cdots, S_h$ during the construction of $T_h$ and the previous trees $T_1, \cdots, T_{h-1}$, each of size at most $s(h)$ is

$$O(h) \cdot s(h) \in O(h^{r+1})s(c) \tag{96}$$

since $s(h) = s((h/c)c) \leqslant (h/c)^r s(c) \leqslant h^r s(c)$ is $r$-log-Lipschitz for a sufficiently large constant $c \geqslant 1$ and $h \geqslant c^2$, by Definition F.2. We now prove $h = O(\log n)$.

We have
$$s(h) \leqslant 2^{r-1}(s(|h-1|) + s(|1-0|)) \leqslant 2^r s(h-1), \tag{97}$$

where the first inequality follows from the fact that $s$ is $r$-log-Lipschitz (see (Braverman et al., 2016, Lemma 6.3)), and the second inequality holds since such a function is non-decreasing by definition, so $s(1) \leqslant s(h-1)$. Hence,

$$|P_h| = 2^{h-1}s(h) \leqslant 2^{h-1} \cdot 2^r \cdot s(h-1) = 2^{r+1} \cdot 2^{h-2}s(h-1) = 2^{r+1}|P_{h-1}|, \quad (98)$$

where the first equality is by Line 4, and the inequality is by equation 97. The value of $h$ is then bounded by

$$\begin{aligned}
h &= \log_2\left(\frac{|P_h|}{s(h)}\right) + 1 \leqslant \log_2\left(\frac{2^{r+1}|P_{h-1}|}{s(h)}\right) + 1 \\
&\leqslant \log_2\left(\frac{2^{r+1}n}{s(h)}\right) + 1 \leqslant \log_2\left(2^{r+1}n\right) + 1 \leqslant (r+1) + \log_2 n + 1 \in O(\log n),
\end{aligned} \quad (99)$$

where the first equality is since $|P_h| = 2^{h-1}s(h)$, and the first inequality is by equation 98.

Plugging equation 99 in equation 96 yields an overall memory as in Claim (iii)

$$O(h) \cdot s(h) \in O(h^{r+1})s(c) \subseteq O(\log n)^{r+1}s(c) = O(\log n)s(c).$$

**The multiplicative approximation error** in the coreset for the nodes of the tree $T_h$ increases by a multiplicative factor of $(1 + \varepsilon/h)$ in each level of the tree $T_h$, by Line 9. We have

$$\varepsilon \leqslant \frac{2\varepsilon}{1 + 2\varepsilon} \leqslant \ln(1 + 2\varepsilon) \quad (100)$$

where the first inequality holds since $\frac{x-1}{x} \leqslant \ln x$ for $x > 0$, and the last inequality holds by the assumption $\varepsilon < 1/2$. Hence,

$$\left(1 + \frac{\varepsilon}{h}\right)^h = \left(\left(1 + \frac{\varepsilon}{h}\right)^{h/\varepsilon}\right)^\varepsilon \leqslant e^\varepsilon \leqslant 1 + 2\varepsilon,$$

where the last inequality is by equation 100. so the coreset $T_h = S_h$ that corresponds to the root is a $(2\varepsilon)$-coreset for $P_h$. Hence, $\bigcup_{i=0}^{h-1} T_i$ is a $(2\varepsilon)$-coreset for $\bigcup_{i=0}^{h-1} P_i$. Similarly, $\bigcup_{i=0}^{h} S_i$ is a $(2\varepsilon)$-coreset for the points that were read from $P_h$. Hence, in Line 13, $C'_n$ is an $\varepsilon$-coreset of a union of $(2\varepsilon)$-coresets, which implies that $C'_n$ is a $(4\varepsilon)$-coreset, as $(1 + 2\varepsilon)(1 + \varepsilon) \leqslant (1 + 4\varepsilon)$ where in the last inequality we use the assumption $\varepsilon \leqslant 1/2$. This proves Claim (i).

**Update time.** In the worst case, the "while" loop in Line 8 is executed for all the $h$ levels of $T_h$. In this case, we construct $O(h)$ coresets, each for input of at most $s(h)$ points whose overall weight is at most $n$, in overall $O(h) \cdot \text{time}(s(h), n, \frac{\varepsilon}{h}, \frac{\delta}{4^h})$ time. In Line 13 we compute a coreset for the union of $O(h)$ coresets, each of size at most $s(h)$, which represents $n$ input points, so their overall size is $O(h)s(h)$ and their construction time is time $(O(h)s(h), n, \varepsilon, \delta)$. Substituting in the last expression, $h \in O(\log n)$ and $O(h) = O(h^{r+1})s(c)$ by equation 99 and equation 96, respectively, yields the update time in Clam (iv), as

$$\begin{aligned}
&O(h)\text{time}\left(s(h), n, \frac{\varepsilon}{h}, \frac{\delta}{4^h}\right) + (O(h)s(h), \varepsilon, \delta) \subseteq O(h)\text{time}\left(O(h)s(h), n, \frac{\varepsilon}{h}, \frac{\delta/4^h}{4^h s(h)}\right) \\
&\subseteq O(\log n)\text{time}\left(s(c)\log^{r+1} n, n, \frac{\varepsilon}{O(\log(n))}, \frac{\delta}{n^{O(1)}}\right).
\end{aligned}$$

**The overall running time** for computing $C'_n$ is obtained by multiplying the update time per point in the previous paragraph by $n$ updates, which proves Claim (v). $\qquad\square$

### F.1 HALVING CALCULUS

If we have a coreset of size $m(\varepsilon, \delta)$ that is independent of the total weight of the or number of input points, for every input set $P$, then $\text{size}(n, \cdot, \varepsilon/h, \delta/h) \leqslant n$ for $n = s(h)$ if $s(h) = m(\varepsilon/h, \delta/h)$. Otherwise the analysis is a bit more involved. In this case we need to solve equations such as $\log(n) = n/2$ for computing the halving function. There is no close solution for such equations but the solution can be represented by the following function which can be computed in a very high precision (a bit for every iteration) using e.g. Newton-Raphson method Chapeau-Blondeau and Monir (2002).

**Lemma F.6** (Lambert $W$ function Barry et al. (2000))**.** *There is a unique monotonic decreasing function $W_{-1} : [-1/e, 0] \to [-1, -\infty)$ that satisfies*

$$x = W_{-1}(x)e^{W_{-1}(x)},$$

*for every $x \in [-1/e, 0]$. It is called the lower branch of the Lambert $W$ function.*

The following lemma would be useful to compute the halving function of coresets whose size depend on $n$.

**Lemma F.7.** *For every $c \geqslant 1$ and $\varepsilon \in (0, 1/e^c]$, if*

$$n \geqslant \left(\frac{4}{\varepsilon} \ln \frac{4}{\varepsilon}\right)^c, \tag{101}$$

*then*

$$n \geqslant \left(\frac{\ln n}{c\varepsilon}\right)^c. \tag{102}$$

*Equality holds for $n = e^{-cW_{-1}(-\varepsilon)}$.*

*Proof.* We denote the Lambert W function by $w$ instead of $W_{-1}$ for simplicity. Multiplying equation 102 by $(c\varepsilon)^c/n$ yields that we need to prove $(c\varepsilon)^c \geqslant (\ln^c n)/n$. The right hand side is a non-increasing monotonic function in the range $n \geqslant 2e^c \ln(2e^c) \geqslant e^c$, as the enumerator of its derivation is

$$(\ln^c n)'n - \ln^c(n) = (c\ln^{c-1} n) \cdot (\ln n)'n - \ln^c n = \ln^{c-1} n(c - \ln n) \leqslant 0.$$

By letting $x = e^{-cw(-\varepsilon)}$, it thus suffices to prove that (i) $(c\varepsilon)^c = \ln^c(x)/x$, and (ii) $n \geqslant x$.

**Proof of (i):** Taking $-(1/c)\ln$ of both sides in $x = e^{-cw(-\varepsilon)}$ yields $-\ln(x)/c = w(-\varepsilon)$. Hence,

$$c\varepsilon = -c(-\varepsilon) = -c \cdot w(-\varepsilon)e^{w(-\varepsilon)} = \ln(x)e^{-\ln(x)/c} = \frac{\ln(x)}{x^{1/c}},$$

where the first equality is by the definition of $w(-\varepsilon)$. Taking the power of $c$ proves **(i)**.

**Proof of (ii):** By equation 101, to prove $n \geqslant x$ it suffices to prove

$$\left(\frac{4}{\varepsilon} \ln \frac{4}{\varepsilon}\right)^c \geqslant x. \tag{103}$$

Let $a = \varepsilon/2$ and

$$b = \ln\left(\frac{1}{a} \ln \frac{1}{a^2}\right).$$

By the assumption $\varepsilon \leqslant 1/e^c \leqslant 1/e$, we have

$$b = \ln\left(\frac{2}{\varepsilon} \ln \frac{4}{\varepsilon^2}\right) \geqslant \ln(2\ln(4)) \geqslant \ln(2e) \geqslant 1. \tag{104}$$

Hence,

$$-a \leqslant -a \cdot \frac{\ln\left(\frac{1}{a} \ln \frac{1}{a^2}\right)}{\ln \frac{1}{a^2}} = -\ln\left(\frac{1}{a} \ln \frac{1}{a^2}\right) \cdot \frac{a}{\ln \frac{1}{a^2}} = (-b) \cdot e^{-b} = w^{-1}(-b), \tag{105}$$

where the inequality holds since $1/a \geqslant \ln(1/a^2)$ for $1/a = 2/\varepsilon \geqslant 1$, and the last equality holds by letting $y = w^{-1}(-b)$ so that $w(y) = -b$ and $y = w(y)e^{w(y)}$ by the definition of $w$. Since $w$ is monotonic decreasing we have by equation 105 that

$$w(-a) \geqslant w(w^{-1}(-b)) = -b. \tag{106}$$

This proves equation 103 as

$$x = e^{-cw(-a)} \leqslant e^{cb} = \left(\frac{2}{a} \ln \frac{1}{a}\right)^c \leqslant \left(\frac{4}{\varepsilon} \ln \frac{4}{\varepsilon}\right)^c,$$

where the first inequality is by equation 106. $\qquad\square$

**Corollary F.8.** *Let $\varepsilon, \delta > 0$, and $u : [0, \infty) \to (1, \infty)$ such that $u(\cdot)$ is $r$-log Lipschitz function for some $r \geqslant 1$. Let $c \geqslant 1$ and $\text{size} : [0, \infty)^4 \to [0, \infty)$ be a function such that*

$$\text{size}(2n, \overline{w}, \varepsilon/h, \delta/4^h) \leqslant \left( \frac{u(h) \ln(\overline{w})}{hc} \right)^c \tag{107}$$

*for every $h, n, \overline{w} \geqslant 1$. Let overloading of $s : [0, \infty) \to [0, \infty)$ be a function such that*

$$s(h) \geqslant \left( 4u(h) \ln(4u(h)) \right)^c \tag{108}$$

*for every $h \geqslant 1$. Then $s$ is an $(\varepsilon, \delta, 2cr)$-halving function of the function $\text{size}$; see Definition F.2.*

*Proof.* Let $h \geqslant 1$, $n = s(h)$, and $\overline{w} = 2^h n$. Then $s$ is a halving function of $\text{size}$ as

$$\text{size}(2n, \overline{w}, \varepsilon/h, \delta/4^h) \leqslant \left( \frac{u(h) \ln(\overline{w})}{hc} \right)^c \tag{109}$$

$$= \left( \frac{u(h) \ln(2^h n)}{hc} \right)^c \tag{110}$$

$$\leqslant \left( \frac{u(h) \ln(n)}{c} \right)^c \leqslant n, \tag{111}$$

where equation 109 is by equation 107, equation 110 is by the definition of $\overline{w}$, equation 111 is since $2 \leqslant 4u \leqslant s(h) = n$ by the definition of $s$, and the last inequality holds by replacing $\varepsilon$ with $1/u(h)$ in Lemma F.7.

If $g(x) = (x \ln x)^c$, then for every $x, b \geqslant e > 2$

$$g(bx) = (bx)^c (\ln(b) + \ln x)^c \leqslant (bx)^c (\ln(b) \ln x)^c = (x \ln x)^c (b \ln b)^c$$

$$\leqslant b^{2c} (x \ln x)^c = b^{2c} g(x), \tag{112}$$

where the first inequality holds since $(a + y) \leqslant 2y \leqslant ay$ for every $y \geqslant a \geqslant 2$, and equation 112 holds since $\ln b \leqslant b$. Since $s(h) = g(u(h))$, for every $\Delta \geqslant e$ we have

$$s(\Delta h) = g(u(\Delta h)) \leqslant g(\Delta^r u(h)) \leqslant \Delta^{2cr} g(u(h)) = \Delta^{2cr} s(h),$$

where the first inequality holds since $u$ is $r$-log-Lipschitz and the second holds by substituting $b = \Delta^r$ in equation 112. Jointly with equation 111 we obtain that $s$ is $(\varepsilon, \delta, 2cr)$-halving function of $\text{size}$. $\square$

## G   Wrapping All Together

In this section we use the previous chapter to give an example coreset for any $k$-GMM. First we suggest an inefficient construction (quadratic running time in $n$). Then we use it in the streaming setting to obtain time that is near-linear in $n$.

Michael Edwards and Kasturi R. Varadaraja Edwards and Varadarajan (2005) suggested a coreset for the projective clustering problem where the fitting cost is the maximum distance over every input point to its closest subspace in the query. It was also proven in Edwards and Varadarajan (2005); Har-Peled (2004) that no such coreset of size sub-linear in $n$ exists, unless we assume that the input can be scaled to be on a grid of integers. The suggested coreset size then depends poly-logarithmically on the size of this grid. This assumption is usually reasonable in practice, as, unlike the theoretical RAM model, every coordinate is stored in memory using a small number of bits (e.g. 16 or 32 bits). The original result is for any $\varepsilon' \in (0, 1)$ but due to its usage in Theorem E.7, $\varepsilon' = 1/3$ or any other constant – suffices. The impact on the total sensitivity and thus the overall coreset size would be a factor of $(1 + \varepsilon')$, but the multiplicative approximation error would still be $\varepsilon \in (0, 1)$.

**Theorem G.4** (Projective ClusteringEdwards and Varadarajan (2005) )**.** *Let $M \geqslant 2$ and $k \geqslant 1$ be two integers. Let $g(d, k)$ be a number that depends only on $d$ and $k$, and can be computed from the proof. Let $P \subseteq \{-M, \cdots, M\}^d$, and $C \subseteq P$ be the output of a call to $\ell_\infty$-PROJ-CLUSTERING-CORESET$(P, k, 1/3)$; see Algorithm G.1. Then $C$ is a $(1/3)$-coreset for $(P, H_{d,k}, \text{dist}, \|\cdot\|_\infty)$ of size $(\log M)^{g(d,k)}$. Moreover $C$ can be computed in $n \cdot (\log M)^{O(1) \cdot g(d,k)}$ time.*

---

**Algorithm G.1:** $\ell_\infty$-PROJ-CLUSTERING-CORESET$_{\text{SUBSPACE-CORESET}}(P, k, \varepsilon)$

---

**Input:**      A finite set $P \subseteq \{-M, -M+1, \cdots, M\}^d$ for some integer $M \geqslant 1$,
           an integer $k \geqslant 1$, and an approximation error $\varepsilon \in (0, 1)$.
**Required:** An algorithm SUBSPACE-CORESET$(P, \varepsilon)$ that returns an $\varepsilon$-coreset
           for $(P, H_{d,1}, \text{dist}, \|\cdot\|_\infty)$.
**Output:**    An $\varepsilon$-coreset $C \subseteq P$ for $(P, H_{d,k}, \text{dist}, \|\cdot\|_\infty)$; see Theorem G.4.

1 **if** $k = 1$ **then**
2     **return** SUBSPACE-CORESET$(P, \varepsilon)$
3 $C_0 := \ell_\infty$-PROJ-CLUSTERING-CORESET$(P, k-1, \varepsilon)$
4 **for** *every* $v_0 \in C_0$ **do**
5     $P[v_0] := P$
6     $C[v_0] := $ RECURSIVE$(P[v_0], k, \varepsilon, \{v_0\})$; see Algorithm G.2
7     $C_1 := C \cup C[v_0]$
8 $C := C_0 \cup C_1$
9 **return** $C$

---

---

**Algorithm G.2:** RECURSION$(P, k, \varepsilon, V)$

---

**Input:**      A finite set $P \subseteq \{-M, -M+1, \cdots, M\}^d$ for some integer $M \geqslant 1$,
           an integer $k \geqslant 1$, an approximation error $\varepsilon \in (0, 1]$, and a set $V = \{v_0, \cdots, v_t\} \subseteq P$.
**Output:** A set $C \subseteq P$.

1 $C := \varnothing$; $A[\{v_0\}] := \{v_0\}$
2 **if** $t \geqslant 1$ **then**
3     **for** $i := 1$ *to* $t$ **do**
4        $A[\{v_0, \cdots, v_i\}] := \left\{ \sum_{h=0}^i \alpha_h v_h \mid (\alpha_0, \cdots, \alpha_i) \in \mathbb{R}^{i+1}, \sum_{b=0}^i \alpha_b = 1 \right\}$
         `// The affine i-subspace that passes through` $v_0, \cdots, v_i$.
5        Set $\pi(v_i, A[\{v_0, \cdots, v_{i-1}\}]) \in \arg\min_{x \in A[\{v_0, \cdots, v_{i-1}\}]} \|v_i - x\|_2$
         `// The projection (closest point) of` $v_i$ `onto` $A[\{v_0, \cdots, v_{i-1}\}]$
6        $u_i := v_i - \pi(v_i, A[\{v_0, \cdots, v_{i-1}\}])$
         `// The vector from` $v_i$ `to its projection on` $A[\{v_0, \cdots, v_{i-1}\}]$
7     $R[V] := \{v_0 + a_1 u_1 + \cdots + a_t u_t \mid a_i \in [-1, 1], i \in [t]\}$ `// A` $t$`-dimensional`
       `rectangle centered at` $v_0$ `whose` $i$`th side length is` $2\|u_i\|$.
8     $r[V] := \{a_1 u_1 + \cdots + a_t u_t \mid a_i \in [-\varepsilon, \varepsilon], i \in [t]\}$
       `// A` $t$`-dimensional rectangle whose` $i$`th side length is` $2\varepsilon\|u_i\|$
9     $\mathcal{R}[V] := $ A partition of $R[V]$ into $\frac{1}{\varepsilon^t}$ translated copies of $r[V]$
10    **for** *each rectangle* $R \in \mathcal{R}[V]$ **do**
11       $C_R[V] := \ell_\infty$-PROJ-CLUSTERING-CORESET$(P \cap R, k-1, \varepsilon)$
       `// see Algorithm G.1.`
12      $C := C \cup C_R[V]$
13 **if** $t \leqslant d - 1$ **then**
14    $B_0[V] := P \cap A[V]$
15    $c := 1/d^{3(d+1)/2}$
16    **for** $j := 1$ *to* $8d \log_2 M + \log_2(1/c)$ **do**
17      $B_j[V] := \left\{ p \in P \mid 2^{j-1} c/M^{d+1} \leqslant \text{dist}(p, A[V]) < 2^j c/M^{d+1} \right\}$
      `// dist(p, A[V]) is the distance from` $p$ `to` $A[V]$.
18      $K_j[V] := \ell_\infty$-PROJ-CLUSTERING-CORESET$(B_j[V], k-1, \varepsilon)$
19      $C := C \cup K_j[V]$
20      **for** *every* $v_{t+1} \in K_j[V]$ **do**
21        $V' := V \cup \{v_{t+1}\}$    `//` $V' = \{v_0, \cdots, v_{t+1}\}$
22        $P[V'] := \bigcup_{i=0}^j B_i[V]$
23        $C[V'] := $ RECURSION$(P[V'], k-1, \varepsilon, V')$
24        $C := C \cup C[V']$
25 **return** $C$

---

---

**Algorithm G.3:** $k$-GMM-CORESET$(P', k, \varepsilon, \delta)$

---

**Input:** A weighted set $P' = (P, w)$ of points in $\mathbb{R}^d$, $k \in \mathbb{N} \cap [1, \infty]$ number of clusters,
an approximation error $\varepsilon > 0$ and a probability $\delta$ of failure
**Output:** $\varepsilon$-coreset $(C, u)$ for $k$-GMMs of $P$, with probability at least $1 - \delta$; see Theorem G.5.

1 $\ell_\infty \text{ALG} := \ell_\infty$-CORESET$(P, k, \varepsilon)$ // See Algorithm G.1
2 $s := \text{WSENSITIVITY}_{\ell_\infty}$-CORESET$(P', \varepsilon, \delta, \ell_\infty \text{ALG})$ // See Algorithm E.4
3 $t := \sum_{p \in P} s(p)$
4 $d' = k^4 d^3$
5 $m := f(\varepsilon, \delta, d', t)$
6 $(C, u) := \text{CORESET}(P, w, s, m)$ // See Algorithm B.6
7 **Return** $(C, u)$

---

The following lemma states our main application which is an $\varepsilon$-coreset that approximates the sum $\|\cdot\|_1$ of fitting error $l$ for *any* $k$-GMM. We apply the reduction from $\|\cdot\|_\infty$ to $\|\cdot\|_1$ from the previous chapters. The running time is quadratic in $n$ but would be reduced to linear in Section F by applying it only on small weighted subsets of $P$ in Theorem G.6. This is also the reason why the lemma is stated for weighted input.

**Lemma G.5.** *Let $M \geqslant 2$ and $P' = (P, w)$ be a weighted set of $n := |P|$ points in $\{-M, -M + 1, \cdots, M\}^d$. Let $g(d, k)$ be a number that depends only on $d$ and $k$, as defined in Theorem G.4. Let $\varepsilon, \delta \in (0, 1/10)$, and $(C, u)$ be the output of a call to $K$-GMM-CORESET$(P', k, \varepsilon, \delta)$; see Algorithm G.3.*

*Then, with probability at least $1 - \delta$, $(C, u)$ is an $\varepsilon$-coreset for $(P', \Theta_{d,k}, L, \|\cdot\|_1)$, where*

$$|C| \leqslant (\log M)^{g(d,k)} \cdot \frac{\log^2 \overline{w}(P')}{\varepsilon^2} \log\left(\frac{1}{\delta}\right),$$

*its computation time is*

$$O(n^2) \cdot (\log M)^{g(d,k)},$$

*and $\sum_{p \in C} u(p) = \sum_{p \in P} w(p)$.*

*Proof.* Let

$$D := \left\{ (p; 0; \cdots; 0) \in \mathbb{R}^{2d+1} \mid p \in P \right\}.$$

By substituting $C := S$ in Theorem G.4, a $(1/3)$-coreset $S'$ for $(D, H_{2d+1,k}, \text{dist}, \|\cdot\|_\infty)$ of size $|S'| \in (\log M)^{g(d,k)}$ can be computed in time $n \cdot |S'|^{O(1)}$. Therefore, $S = \{p \in P \mid (p; 0; \cdots; 0) \in S'\}$ is an $O(k)$-coreset for $(P, \Theta_{d,k}, L, \|\cdot\|_\infty)$; see formal proof in Theorem G.7 Therefore, $(\ell_\infty\text{-CORESET}, \text{size}, \text{time})$ is an $\varepsilon'$-coreset scheme for $(P, \Theta_{d,k}, L, \|\cdot\|_\infty)$ where $\varepsilon' \in O(k)$, $\text{size}(\cdot, \cdot, \varepsilon', \cdot) = |S| = |S'|$ and $\text{time}(n, \cdot, \varepsilon', \cdot) \in n \cdot |S'|^{O(1)}$.

Substituting $\varepsilon = \varepsilon'$, $\delta = 0$, $\text{cost} := L$, and $Y = \Theta_{d,k}$ in Theorem E.7 yields that we can compute a sensitivity bound $s : P \to [0, \infty)$ for $(P', \Theta_{d,k}, L)$, whose total sensitivity is

$$t = \sum_{p \in P} s(p) \in \text{size}(n, n, \varepsilon', 0) \cdot O\left(\log \frac{\overline{w}(P')}{\varepsilon'}\right) \subseteq |S'| \cdot O\left(\log \overline{w}(P')\right) = (\log M)^{O(1)} \cdot O\left(\log \overline{w}(P')\right),$$

in time $n \cdot |S|^{O(1)}$.

By Corollary A.4, the dimension of $(P, \Theta_{d,k}, f)$ is $d' \in O(d^2 k^4)$ where $f$ is defined there in equation 43. Plugging $s$ in Corollary E.8 and choosing $\varepsilon, \delta \in (0, 1)$ yields that an $\varepsilon$-coreset $(C, u)$ for $(P', \Theta_{d,k}, L, \|\cdot\|_1)$ of size

$$|C| \in O(1) \cdot \frac{(t+1)}{\varepsilon^2}\left(d' \log(t+1) + \log\left(\frac{1}{\delta}\right)\right)$$

$$= O(1) \cdot \frac{(t+1)}{\varepsilon^2}\left(d^2 k^4 \log(t+1) + \log\left(\frac{1}{\delta}\right)\right)$$

$$\subseteq O(1) \cdot \left(\frac{t}{\varepsilon}\right)^2 \log\left(\frac{1}{\delta}\right)$$

can be computed in $O(n) \cdot n \cdot |S|^{O(1)} = O(n^2) \cdot (\log M)^{g(d,k)}$ time, with probability at least $1 - \delta$. □

The following theorem improve the running time of the previous lemma to be only linear in $n$, via the streaming framework.

**Theorem G.6.** *Let $M \geqslant 2$ be an integer, and $stream$ be a stream of points from $\{-M, -M+1, \cdots, M\}^d$. Let $\varepsilon, \delta \in (0, 1)$, and for every $h \geqslant 1$ let*

$$s(h) = \frac{h}{\varepsilon^2} \log \frac{h}{\varepsilon} \cdot \log^2 \left( \frac{1}{\delta} \right) \log^{2g(d,k)} M, \tag{113}$$

*where $g(d, k)$ is a function that depends only on $d$ and $k$ as defined in Theorem G.4. Let $k$-GMM-CORESET be defined as in Algorithm G.3, and $C'_1, C'_2, \cdots$ be the output of a call to $\text{STREAMING}_{\text{CORESET}}(stream, \frac{\varepsilon}{6}, \frac{\delta}{6}, k\text{-GMM-CORESET}, s)$; see Algorithm F.4. Then, with probability at least $1 - \delta$, the following hold.*

*For every integer $n \geqslant 1$:*

    *(i) (Correctness) $C'_n$ is an $\varepsilon$-coreset of $(P_n, \Theta_{d,k}, L, \|\cdot\|_1)$ and for $(P_n, \Theta_{d,k}, \phi, \|\cdot\|_1)$, where $P_n$ is the first $n$ points in $stream$.*

    *(ii) (Size)*

$$|C_n| \in (\log M)^{g(d,k)} \cdot \frac{\log^2 n}{\varepsilon^2} \log \left( \frac{1}{\delta} \right).$$

    *(iii) (Memory) there are*

$$b \in \frac{1}{\varepsilon^2} \log \frac{1}{\varepsilon} \cdot \log^2 \left( \frac{1}{\delta} \right) \log^{O(1)}(n) \log^{2g(d,k)} M$$

    *points in memory during the streaming.*

    *(iv) (Update time) $C'_n$ is outputted in additional $t \in O(b^2) \cdot (\log M)^{g(d,k)}$ time after $C'_{n-1}$.*

    *(v) (Overall time) $C'_n$ is computed in $nt$ time.*

*Proof.* Let $h, w' \geqslant 1$ and $n = s(h)$. Substituting $P' = P_n$ in Lemma G.5 yields, $(k\text{-GMM-CORESET}, \text{size}, \text{time})$ is an $(\varepsilon, \delta)$-coreset scheme for $(P_n, \Theta_{d,k}, \phi, \|\cdot\|_1)$, where

$$\text{size}(n, w', \varepsilon, \delta) \leqslant (\log M)^{g(d,k)} \cdot \left( \frac{\log(w')}{\varepsilon} \right)^2 \log \left( \frac{1}{\delta} \right), \tag{114}$$

and

$$\text{time}(n, w', \varepsilon, \delta) \in O(n^2) \cdot (\log M)^{g(d,k)}.$$

Let $h \geqslant 1$, and

$$u(h) = \left( \log M^{g(d,k)} \cdot \frac{4h^5}{\varepsilon^2} \cdot \log \left( \frac{4}{\delta} \right) \right)^{1/2}. \tag{115}$$

Hence,

$$\text{size}(2n, w', \varepsilon/h, \delta/4^h) \leqslant (\log M)^{g(d,k)} \cdot \left( \frac{h \log(w')}{\varepsilon} \right)^2 \log \left( \frac{4^h}{\delta} \right) \leqslant \left( \frac{u(h) \log(w')}{4h} \right)^2,$$

where the first inequality is by equation 114. Since $u$ is $(5/2)$-log-Lipschitz, and

$$s(h) \geqslant 10u^3(h) \geqslant (4u(h) \ln 4u(h))^2,$$

by equation 113 and equation 115, substituting $r = (5/2)$ and $c = 2$ in Corollary F.8 yields that $s$ is $(\varepsilon, \delta, 15)$-halving of size. Substituting

$$s(24) \in \left( \log^{g(d,k)} M \log \left( \frac{1}{\delta} \right) \right)^2 \cdot \frac{1}{\varepsilon^2} \log \frac{1}{\varepsilon},$$

and

$$\text{time}(b, n, \frac{\varepsilon}{O(\log n)}, \frac{\delta}{n^{O(1)}}) \in O(b^2) \cdot (\log M)^{g(d,k)}$$

in Theorem F.5 then proves Theorem G.6 for the query space $(P_n, \Theta_{d,k}, \phi, \|\cdot\|_1)$. Substituting $P := P_n$ in Observation C.1, proves the theorem also for $(P_n, \Theta_{d,k}, L, \|\cdot\|_1)$. $\square$

The following theorem proves a reduction from coreset of projective clustering, to $\|\cdot\|$-coreset for $k$-GMMs based on previous chapters. This coreset might be of independent interest.

**Theorem G.7.** *Let $P$ be a finite set of points in $\mathbb{R}^d$, and $k \geq 1$ be an integer. Let $D := \{(p; 0; \cdots ; 0) \in \mathbb{R}^{2d+1} \mid p \in P\}$, and $C$ be a $(1/3)$-coreset for $(D, H_{2d+1,k}, \text{dist}, \|\cdot\|_\infty)$. Then, $S = \{p \in P \mid (p; 0; \cdots ; 0) \in C\}$ is an $O(k)$-coreset for $(P, \Omega_{d,k}, L, \|\cdot\|_\infty)$.*

*Proof.* Let $\theta \in \Omega_{d,k}$. By Lemma D.1, there is a $k$-SMM $y \in Y_{2d+1,k}$ that satisfies

$$\phi(p, \theta) = \text{cost}(x, y), \tag{116}$$

for every $p \in P$ and its corresponding point $x = (p; 0; \cdots ; 0) \in P$. By Summing equation 116 over every $p \in P$ and $p \in S$, respectively, we obtain

$$\phi(P, \theta) = \text{cost}_\infty(D, y) \text{ and } \phi(S, \theta) = \text{cost}_\infty(C, y). \tag{117}$$

By Lemma 6.3, $C$ is an $O(k)$-coreset for $(P, Y_{2d+1,k}, \text{cost}, \|\cdot\|_\infty)$. Hence,

$$\text{cost}_\infty(D, y) \in O(k)\text{cost}_\infty(C, y).$$

Combining the last equality with equation 117 yields

$$\phi(P, \theta) = \text{cost}_\infty(D, y) \in O(k)\text{cost}_\infty(C, y) = O(k)\phi(S, \theta).$$

Since $\theta \in \Theta_{d,k}$ was arbitrary, we conclude that $S$ is an $O(k)$-coreset for $(P, \Theta_{d,k}, \phi, \|\cdot\|_\infty)$. By Lemma C.1, $S$ is thus also an $\varepsilon$-coreset for $(P, \Theta_{d,k}, L, \|\cdot\|_\infty)$. $\square$

