# OpenReview forum: "Coresets for Mixtures of (arbitrarily large) Gaussians"
_ICLR.cc/2026/Conference — Submitted to ICLR 2026_

### Official Review · Reviewer_ScbR · 2025-10-26

**Soundness:** 3
**Presentation:** 2
**Contribution:** 3
**Rating:** 4
**Confidence:** 4

**Summary:**

The paper designs coresets for k-GMM (a mixture model of k gaussians) in $R^d$. Recall that k-GMM basically defines a probability distribution, and a classical problem is to find the k-GMM that best fits an input dataset P, i.e., find model parameters $\theta$ that maximize the likelihood of P. This problem can be seen as a generalization of k-means.

The paper designs a small coreset C, which is a reweighted subset of the data P, such that the log-likelihood function $L(P,\theta)$ is approximated i.e. by $L(C,\theta)$ within factor $1\pm\epsilon$. Coresets are very useful and have many applications, for example they can reduce storage and communication, and can speed up finding good model parameters $\theta$. The size of this coreset is $polylog(n)/\epsilon^2$, for $n=|P|$, assuming that the parameters $k,d$ are fixed, and that each coordinate has fixed precision (represented by a fixed number of bits). Moreover, the coreset can be constructed in near-linear time using the (standard) merge-and-reduce approach.

Besides this theoretical contribution, the paper also provides experiments to validate its algorithmic approach and compare it to previous work.

**Strengths:**

S1. The k-GMM problem is important, and also designing a small coreset for it.
S2. The paper addresses a family of k-GMMs that is significantly richer than the only prior work on this problem by Lucic et al.: The prior work only handled "semi-spherical" k-GMMs, meaning that each covariance matrix has all its eigenvalues in a bounded range, like $[\epsilon,1]$. The paper requires only the upper bound $1$, which is some sort of normalization.
S3. The methodology here differs from that prior work of Lucic et al. and I think identifies a much better way(the right one?) to tackle this problem

**Weaknesses:**

W1. The result is new but its technical details are somewhat complicated. It builds upon a lot of prior work that has to be adapted (extended to a more general setting) and thus cannot be used in a black-box manner. Besides hampering readability, it is hard to extract which steps have new ideas.
W2. The comparison with prior work by Lucic et al. is incomplete. The submission does not mention the coreset size in that prior work, and I had to find myself in Theorem 2 therein, that their coreset size is $poly(k,d)/\epsilon^2)$, which is independent of $n=|P|$. In this sense, prior work is better, i.e., has smaller coreset size than the submission's coreset size $polylog(n)/\epsilon^2$, with dependence on k and d omitted (so I cannot even be sure it is polynomial and not exponential).
W3. The scheme follows several known methodologies, including sensitivity sampling, a form of importance sampling that is a common approach to construct coresets, and a method of Varadarajan and Xiao to estimate sensitivities of points via a low-quality coreset. In honesty, this is a rather minor weakness, obtaining the results is quite not trivial and in fact requires a lot of effort.
W4. The paper is written in a rather technical way. While obvious right from the first paragraph, this is particularly glaring in section 5, which dumps a lot of technical terms and is overall non-informative.
W5. The experiments are limited and explained in a hap-hazard manner and I did not understand much. See my questions below for details.

**Questions:**

Q1. How does your coreset size depend on d and k? Is it polynomial at all?
- How does it compare to Lucic et al?
- Can you get some bound on the coreset size, even existentially and not efficiently, by a naive adaptation of the very basic approach of [Har-Peled and Mazumdar 2004], at least when in the semi-spherical case which might be similar to k-means?
- Are there any known lower bounds or other reasons to believe your bound is close to optimal, e.g., a logarithmic dependence on n is unavoidable?

Q2. In section 7 (experiments):
- why take a difference between two log-likelihood functions? I suspect it represents additive error of log-likelihood, which corresponds to multiplicative error in the likelihood itself, but you call it "optimal log-likelihood ".
- what is the difference between $G_{org}$ and $G_{trg}$, which are both trained from P?
- in fact, where your new algorithm is used in all these experiments? does it affect downstream applications?
- the last paragraph reads as cherry-picking the "best" places, rather than evaluating all the experiments. What is your conclusion from the entire set of experiments?

Minor suggestions (typos etc., not questions to respond):
- L 136: I don't see any dependency on k (that you say is unavoidable)
- L 149: why assume $\epsilon$ is constant if you write the precise dependency on it?
- L 165: not clear what is $|| \infty ||$
- Fig 1: font is too small to read
- L 202: this paragraph has english problems e.g. change implies to means
- L 232: writing big O for 1/3 is awkward
- L 277: notation $dist_\infty$ was not defined yet, only in (9)
- eqn (8): the -1 inside exp looks suspicious, what is its intuitive explanation and how do you handle it?
- L306: why introduce the letter $\xi=1$
- L 478: change serious to series
- references: some repetitions like [Feldman and Langberg 2011] and [Varadarajan and Xiao 2012].

---

### Official Review · Reviewer_bmYK · 2025-10-31

**Soundness:** 4
**Presentation:** 3
**Contribution:** 2
**Rating:** 4
**Confidence:** 3

**Summary:**

This paper presents a coreset construction for Gaussian Mixture Models. Previous constructions had certain shape requirements that this one removes. There are nice ideas in this paper and the construction seems sound. Even defining what a coreset is precisely for GMMs is somewhat subtle, and the paper does a nice job of that. There are some experiments to demonstrate the feasibility of the approach.

Nonetheless, in my opinion the bar for publishing these types of results has gone up in light of the fact that AI is proving theorems, writing code, designing algorithms, and reviewing papers.

I'm afraid the ICLR audience would find other contributions more relevant to today's developments. This paper might be better suited for a computational geometry conference.

**Strengths:**

The paper does a good job of defining and constructing coresets for GMMs. The construction is nontrivial and the experiments are more extensive than a typical theory paper.

**Weaknesses:**

I'm afraid this paper is somewhat niche for ICLR, especially given the rapid advancements of language models which is driving a lot of interest. While this paper might have been of broad interest ten years ago, I am afraid that it is not the case today.

**Questions:**

Section 1 constrains $\Sigma_i$ to have eigenvalues $\le 1$, while it seems like Lemma C.1 proves $Z(\theta) \le 1$ by assuming $\Sigma_i$​ has eigenvalues $\ge 1$?

---

### Official Review · Reviewer_bxL2 · 2025-11-04

**Soundness:** 2
**Presentation:** 3
**Contribution:** 3
**Rating:** 6
**Confidence:** 3

**Summary:**

This paper provides coreset constructions for Gaussian mixture models of k Gaussians in d-dimensions for constant k,d.  These subsets of input data, if fit with a Gaussian mixture, will (1+$\epsilon$)-approximate the negative-log-likelihood of the cost of the full data.
The result is a coreset of size

  $s = O((1/\epsilon^2) * \log^{g(d,k)} n)$ size

in time O(n s), where g(d,k) is an unspecified function of d,k.

Prior work exists with similar bounds (JMLR 2018), which required control on the eigenvalues of the fit covariance matrices.  This paper does not require such bounds.

It also has some light experiments which shows an empirical improvement in coreset size on the JMLR 2018 paper's method -- on the same datasets they considered.

**Strengths:**

- this is a theory paper, with carefully stated bounds and proofs.  For better or worse, it has a long appendix with detailed proofs.  It also tries to place main new calculations in the paper before the appendix, but it is very dense.  The paper does make a valiant attempt to communicate the ideas in this paper (but it is a lot, and will be hard for those unfamiliar with it)

  - the empirical demonstration of improvement on the JMLR 18 paper is good.

  - the identification, and addressing, of the bounded eigenvalue issue is nice to resolve.

  - the GMM problem is a central one in ML, so worth understanding.

  - the connection to k-subspace projection clustering is a nice way to handle the extreme cases left open in prior work.

**Weaknesses:**

- the improvements provided here are quite theoretical.  For instance, the exponent g(d,k) is not specified, even in the appendix.  It's not clear what role such theory plays towards guiding practice.

  - someone reading this paper, trying to find the algorithm (and caring less about the proof) will have a hard time untangling it from the writing.

  - the experiments are on coreset size, not on computation time.  And does not compare to direct approximation algorithms for GMM.

  - The proof seems to rely on the Braverman-Feldman-Lang-2016 paper on the arxiv.  This paper was never published, and at least some versions seemed to have some technical issues.  Since this paper may only have saved a log factor off of the older bounds, and I think other papers published since (e.g., the Feldman-Schmidt-Sohler SIAM 2020 paper) may have gotten around these issues.  I would prefer if the paper relied on published results, or weaker original results.  I guess the log factors may not make a difference, but I was not able to verify how this aspect of the proof tied in with all aspects of the broader argument.  It's ok to still cite the Braverman-Feldman-Lang-2016 paper for definitions if the authors wish (that paper does indeed make clear some aspects).

**Questions:**

Please address if the claims in the Braverman-Feldman-Lang-2016 are essential to the result, and if so which results and how, or if the bounds the same as those stated can be obtained with weaker results, or those from other published papers.

---

> ### Author Response · Authors · 2025-11-27
> **Answer to Reviewer**
>
> We thank the reviewer for the helpful comments and suggestions.
> While we could have sent this paper to a theoretical paper, our aim was to have impact also on practitioners as discussed by the reviewer.
> Since the submission, we spent a lot of time toward this direction, and hope that the reviewer will increase its scoring.
> Otherwise, we would appreciate if you can tell us what else is missing.
>
> Q1: the exponent $g(d,k)$ is not specified, even in the appendix. It's not clear what role such theory plays towards guiding practice.
> A1: Following this comment, we added the following discussion  to "our contribution":
> "The function $g(d,k)$ in our main Theorem is from Theorem G.4 which is Theorem 1 in [Michael Edwards and Kasturi R. Varadarajan. No coreset, no cry: II.], after replacing $g(d,k)$ with $f(d,k)$. It is a polynomial in $d$ of degree $k$, which corresponds to the recursion in $k$ and $d$ in the pseudo code. In particular, it is constant, i.e., $g(d,k)\in O(1)$ if $d$
>  and $k$ are constants. The exact analysis can be found in Section "Running Time and Size" of the above paper.
>
> Q2: someone reading this paper, trying to find the algorithm (and caring less about the proof) will have a hard time untangling it from the writing.
> A2: We did our best, but expect that the open code will be more useful and accesible for most of the readers.
>
> Q3: the experiments are on coreset size,
> A3: The running time are quite neglected compared to running on the full data, but we added them, including experiments that compare the quality of the results for the same running time.
>
> Q4: "The proof seems to rely on the Braverman-Feldman-Lang-2016 p". Please address if the claims in the Braverman-Feldman-Lang-2016 are essential to the result,
>
> A4: The paper uses only the sensitivity framework as suggested by Langberg and Feldman. The paper of Braverman et al. mainly focus on improvements for k-means. We remove all the references to the mentioned arxiv version.

---

### Official Review · Reviewer_UgsC · 2025-11-04

**Soundness:** 2
**Presentation:** 2
**Contribution:** 2
**Rating:** 4
**Confidence:** 3

**Summary:**

The paper proposes coresets for the $k$-GMM problem with arbitrary covariance matrices. The overall coreset construction framework is the standard sensitivity-based sampling framework, however, to calculate the bounds on the sensitivity scores, the paper uses reduction of the problem first to what the authors call $k$-SMM (subspace mixture model) and then to projective clustering. They support their theoretical results with a small set of experiments.

**Strengths:**

1) The problem is well motivated and will be of interest to the community.
2) The overall idea of combining different blocks from the coreset literature and the reduction of coreset for arbitrary $k$-GMM to the problem of coreset for projective clustering is interesting.
3) The use of Figure 1 (I have some issue with that. See weaknesses) and Figure 2 overall add to the readability of the paper and I appreciate it.
4) Some empirical evidence of the effectiveness of the algorithm is provided.

**Weaknesses:**

Main issue with the paper is of writing and presentation. The paper reads good till section 3. The problem is described and motivated very clearly. However, the quality and clarity of writing go downhill after that. I understand the space constraints. However, this still requires improvement. There are quite a few typos and notations that create confusion and do not allow to verify proofs easily. I will list down some specific examples below:
1) On line 123 "there are very little results" must be "few results". This does not read natural.
2) line 138 "generalized to sum of distances" and not "generalized sum of distances", if I understand correctly
3) line 146 w should be uppercase.
4) line 160 Generalization title on one page and then content shifts to next page again looks unnatural. Also why do the authors say "folklore" results is not clear as most of the mentioned results are proved.
5) In figure 1 should not the direction of the arrows be reversed as you first build coresets for the projective clustering and not the other way round? Or are the authors trying to describe the flow of the paper?
6) Use of $\mathcal{Q} $ and $P$ interchangeably to describe the four tuple in the beginning of section 5 is confusing. $\mathcal{Q}$ is typically used for query space in coreset literature.
7) Line 290 -293, containing equation (9) defines cost which is max however talks about point in $P$ with maximum cost which must be argmax.
8) In Lemma 6.3. $C$ is coreset for $H_{d,k}$ (hyperplanes), then why does eq.11 hold true directly for $Y_{d,k} $(subspaces) ? Does it mean each $S_i$ being considered in $y$ is formed by the $k$-Hyperplanes? Or am I missing something?
I think all these, and other such issues need to be resolved before the paper becoming of acceptable quality at a venue like ICLR.

Other weakness of the paper is that the experiments are very few, the authors should also try to evaluate the coresets on the basis of time taken for construction and training, do their coresets also work on some arbitrary synthetic covariance matrices as that is the main contribution of the paper.

**Questions:**

See weaknesses

---

> ### Author Response · Authors · 2025-11-27
>
> We thank the reviewer  for the careful reading that is not so common these days.
> Most of the comments were justifies, but also easy to fix.
> In the recent days we worked hard to add more experiments and really hope that the updates below will convince the reviewer to raise the score. Otherwise, please advice us what is still missing.
>
> Answer to questions:
> Q1. " why do the authors say "folklore" results is not clear as most of the mentioned results are proved."
> A1.  We added the text:
> "We could not find some of these folklore results in the literature, and others were inaccurate or too specific. To this end, the appendix include full formal proofs for these results and related algorithms, usually with a serious generalization."
>
> Q2. "In figure 1 should not the direction of the arrows be reversed "
> A2. For clarification, the following text was added below the figure:
> "The arrows points the direction of reduction, from the original problems (that are closer to the roots of the tree) to the problems in the bottom of the tree, which are easier to solve or have known solutions. This is also similar 	to the flow of the paper."
>
> Q3. "Use of $Q$  and $P$ was used  interchangeably."
> A3. We did not want to overload $P$, but $Q$ with a different font than the query space may indeed be confusing.
> We change $Q$ to $P'$, as in the algorithms.
>
> Q4. "Defines cost which is max however talks about point  with maximum cost"
> A4. Indeed. The text was changed to "be the distance of the point in P with the maximum distance..."
>
> Q5. "why does eq.11 hold true directly for (subspaces) ? "
> A5. Since the coresets for hyperplanes (d-1 subspaces) also holds for k-subspaces where the dimension is less than d-1, as a special case. We added a simple proof.
>
> Q6. The experiments are very few,
> A6. Following the suggestions of the reviewer, we added more evaluations, especially experiments for some synthetic covariance matrices. The results are actually much better than the in the previous version, and demonstrates how the improvement factor approaches infinity compared to existing heuristics and [1].

---

### Official Review · Reviewer_4QbS · 2025-11-07

**Soundness:** 3
**Presentation:** 2
**Contribution:** 2
**Rating:** 6
**Confidence:** 3

**Summary:**

This paper proposes the first randomized algorithm to construct $\epsilon$-coresets for general k-Gaussian Mixture Models (k-GMMs), extending beyond previous work that can only handle near-spherical GMMs. The method leverages connections to projective clustering and computational geometry to approximate the negative log-likelihood of any $k$-GMM up to a $(1 + \epsilon)$ factor. Experimental results on real-world datasets validate the proposed approach.

**Strengths:**

The core contribution lies in a theoretical framework that connects coreset construction for $k$-GMMs with projective clustering and classic computational geometry. The authors leverage sensitivity-based importance sampling, merge-reduce tree structures, and streaming algorithms to achieve near-linear time complexity in the number of input points. The resulting coreset approximates the negative log-likelihood of the full dataset for any $k$-GMM model.

**Weaknesses:**

1. The motivation for studying GMM based clustering model is unclear. A more comprehensive discussion on GMM models should be provided.

2. Although the coreset size is stated to be independent of the dimension $d$, the techniques used to eliminate this dependence are not clearly explained.

3. The main proofs in the paper are hard to follow, especially the part that leverages projective clustering to handle general k-GMMs.

4. The proposed framework appears largely built upon existing projective-clustering and coreset-reduction techniques, while the theoretical or methodological novelty is somewhat unclear. It seems that the paper introduces a new “reduction to projective clustering” and defines the notion of k-SMMs, these ideas still closely follow prior formulations.

**Questions:**

1. Can the authors provide some intuitive explanation of how the proposed coreset construction manages to remove the dependence on the data dimension $d$ in the coreset size?

2. The paper claims an $\epsilon$-coreset with near-logarithmic size in $n$ and near-linear construction time. Could the authors provide a clearer intuition or simplified bound showing how the complexity scales with $k$, $d$, and $\epsilon$ (i.e., for the $(\log M)^{g(d, k)}$ term in the coreset size, how to explicitly calculate $g(d, k)$)?

3. Since the theoretical construction is for arbitrary large Gaussians, were any experiments conducted with highly anisotropic covariances to confirm the robustness under non-spherical settings?

4. The experiments on MNIST, Higgs, and CSN datasets show smaller approximation errors, but the setup appears to compare likelihood reconstruction errors rather than downstream GMM performance. Could the authors justify this choice or provide evidence that smaller errors correlate with better model quality?

---

### Official Review · Reviewer_jah5 · 2025-11-08

**Soundness:** 2
**Presentation:** 2
**Contribution:** 2
**Rating:** 4
**Confidence:** 3

**Summary:**

This paper investigates the fundamental machine learning problem of fitting a given dataset with a k-Gaussian Mixture Model (k-GMM). Existing coreset constructions for k-GMMs mainly focus on semi-spherical mixtures, where all covariance matrices are close to the identity matrix, thus limiting their applicability to general, anisotropic Gaussian components. To address these limitations, the paper establishes a new theoretical and algorithmic foundation by introducing the first randomized algorithm that constructs an ε-coreset for arbitrary large-scale k-GMMs. The authors build a link between projective clustering and coreset construction in computational geometry, enabling efficient approximation of the negative log-likelihood for any given model. Furthermore, they demonstrate that the proposed coreset supports downstream tasks such as streaming, distributed, and hyperparameter-calibration settings, all within near-logarithmic time.

**Strengths:**

1. This paper proposes a new reduction from the k-GMM coreset construction problem to projective clustering, offering an elegant geometric foundation that supports efficient ε-coreset computation for arbitrary Gaussian mixtures.

2. Experimental results on real datasets indicate that the proposed method achieves slightly better performance than the two baseline approaches discussed in the paper.

**Weaknesses:**

1. The claimed contributions appear to be mainly a combination or integration of the methods from [1] and [2], lacking sufficient novelty or independent contribution. The authors are encouraged to further clarify the fundamental differences and innovative aspects of their approach compared to these two works.

2. The proposed method seems very similar to the algorithmic framework and technical pipeline described in [2]. It is recommended that [2] be included as a baseline in the experimental comparison to more convincingly demonstrate the improvements and advantages of the proposed approach.

3. The paper’s organization could be improved, and there are major formatting problems in the references section.

[1] Dan Feldman, Matthew Faulkner, and Andreas Krause. Scalable training of mixture models via coresets. In Advances in neural information processing systems, pages2142–2150,2011.

[2] Feldman, Dan, Zahi Kfir, and Xuan Wu. Coresets for gaussian mixture models of any shape. arXiv preprint arXiv:1906.04895 (2019).

**Questions:**

1. The claimed contributions appear to combine the ideas from [1] and [2]. Could the authors clarify what distinguishes their approach from these two works and explain its independent novelty?

2. The proposed algorithm seems closely related to the method described in [2]. Why was [2] not included as a baseline in the experimental comparison?

---

### Meta-Review · Area_Chair_cCek · 2026-01-04

**Summary:**

This is a theory paper that focuses on improving the corset size bound of k-GMM. This work improves upon prior work JMLR 2018. The prior work requires control on the eigenvalues of the fit covariance matrices. This work relaxed that condition.


Strengths:

The reviewers all acknowledged the fundamental nature of finding corset of Gaussian mixture models. They also praised the mathematical rigor of the work. They also pointed out that this work has more experiments than usual theory papers (although the overall experiment is light).

Weaknesses:

The reviewers find that the work might be marginally out of scope of ICLR, the focus of which is less on classical statistical learning. The technical contents are too dense to follow. A reviewer questioned the novelty beyond the main baselines [1] and [2] and recommended to include [2] as a baseline. A reviewer also feels that the used methodologies are from existing literature, and combining them makes novelty unclear.

This is a heavy theory paper that studies a classical problem in statistical learning. The problem itself is fundamental and thus could be of interest. However, the nature of the paper does not seem to be a good fit to ICLR that focuses on representation learning. The relatively short review process may not be able to offer a thorough examination for the lengthy and heavy proof of the work. Some reviewers have concerns on the innovation of the proof techniques used in this work, but this question was not answered.

**Reviewer Concerns:**

The rebuttal added some additional experiments, but does not address most of the weaknesses (only responded to 2 out of 6 reviewers).

**Reviewer Scores:**

The rebuttal added some additional experiments, but does not address most of the weaknesses (only responded to 2 out of 6 reviewers).

The rebuttal answered notation and clarity questions. But these do not change the nature of the work that perhaps is out of the scope of ICLR.

---

### Decision · Program_Chairs · 2026-01-26

Reject